# Language Imbalance Driven Rewarding for Multilingual Self-improving

**Wen Yang[1,2]** [*] **, Junhong Wu[1,2]** [*] **Chen Wang[1,2], Chengqing Zong[1,2], Jiajun Zhang[1,2,3,4]** [†]

[1] School of Artificial Intelligence, University of Chinese Academy of Sciences
[2] Institute of Automation, Chinese Academy of Sciences
[3] Wuhan AI Research  [4] Shanghai Artificial Intelligence Laboratory, Shanghai, China
{yangwen2023, wujunhong2021, wangchen2020}@ia.ac.cn
{cqzong, jjzhang}@nlpr.ia.ac.cn

## Abstract

Large Language Models (LLMs) have achieved state-of-the-art performance across numerous tasks. However, these advancements have predominantly benefited "first-class" languages such as English and Chinese, leaving many other languages underrepresented. This imbalance, while limiting broader applications, generates a natural preference ranking between languages, offering an opportunity to bootstrap the multilingual capabilities of LLM in a self-improving manner. Thus, we propose *Language Imbalance Driven Rewarding*, where the inherent imbalance between dominant and non-dominant languages within LLMs is leveraged as a reward signal. Iterative DPO training demonstrates that this approach not only enhances LLM performance in non-dominant languages but also improves the dominant language's capacity, thereby yielding an iterative reward signal. Fine-tuning Meta-Llama-3-8B-Instruct over two iterations of this approach results in continuous improvements in multilingual performance across instruction-following and arithmetic reasoning tasks, evidenced by an average improvement of 7.46% win rate on the X-AlpacaEval leaderboard and 13.9% accuracy on the MGSM benchmark. This work serves as an initial exploration, paving the way for multilingual self-improvement of LLMs. The code is available at `https://github.com/ZNLP/Language-Imbalance-Driven-Rewarding`

## 1 Introduction

Large Language Models (LLMs) have revolutionized the field of Natural Language Processing (NLP) with superior performance across numerous tasks. However, existing studies show that due to the imbalance of pre-training and fine-tuning data across languages, existing LLMs have predominately benefited a few "first-class" languages, particularly *English* and *Chinese*, thereby overlooking a wide range of other languages (Qin et al., 2024). Given that LLMs are used worldwide, such language imbalance presents significant risks for users who operate in less dominant languages (Deshpande et al., 2023). To this end, enhancing the multilingual performance of LLMs has gained increasing attention.

Previous research predominantly frames this imbalance as an issue to be resolved, often addressing it through multilingual training and cross-lingual alignment. The first approach aims to improve multilingual performance by incorporating additional multilingual data (Wei et al., 2023; Dang et al., 2024). However, high-quality multilingual instruction tuning and preference data, particularly for low-resource languages, remain scarce and expensive (Boubdir et al., 2023; Chaudhari et al., 2024). The second approach seeks to bridge the performance gap between languages by aligning non-dominant and dominant ones (Li et al., 2023a; Chen et al., 2023b; Chai et al., 2024; Zhang et al., 2024), which are often bottlenecked by the performance of the dominant language.

---

[*] Equal contribution
[†] Corresponding author

This work takes a different perspective, positing that language imbalance, while still an issue, *creates a natural preference ranking between dominant and non-dominant languages*, which can be leveraged as a reward signal. As the preference ranking is mutual, the reward signal benefits both dominant and non-dominant languages, enabling their simultaneous optimization. Consequently, reliance on human-authored data is eliminated, and the performance ceiling for dominant languages is surpassed.

We thus introduce *Language Imbalance Driven Rewarding*, which leverages the reward generated from inherent language imbalance to enhance the multilingual capabilities of LLM in a self-improving manner. Specifically, our approach adopts an Iterative Direct Preference Optimization (DPO) (Rafailov et al., 2024) similar to previous works (Yuan et al., 2024). As shown in Figure 1, starting from any instruction model with basic multilingual capabilities, responses are generated by the model for multilingual prompts and are then mutually translated by that same model. This translation process largely preserves the original preference rankings yielding from language imbalance (discussed in Section 3.2), allowing for the construction of a preference dataset where responses in the dominant language are treated as preferred and those in the non-dominant language as rejected. Subsequently, our approach employs a variant of the DPO that incorporates a negative log-likelihood (NLL) loss term for the chosen labels and has been demonstrated to be crucial for performance in Pang et al. (2024). The DPO training is executed on both dominant and non-dominant languages, enhancing their performance simultaneously. The model trained with DPO is capable of continuously providing reward signals in proceeding iterations.

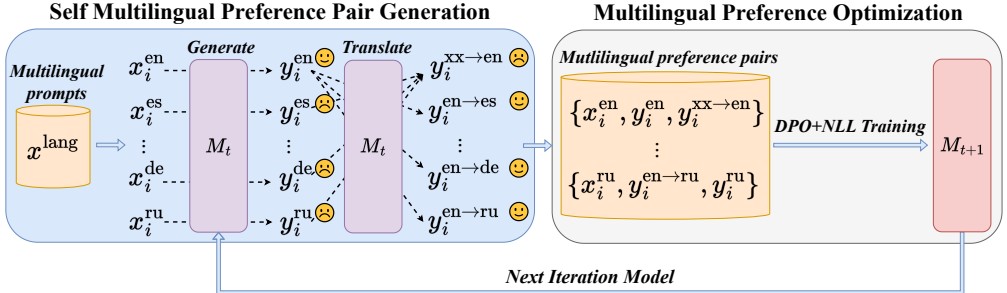

Figure 1: **Language Imbalance Driven Rewarding.** Our method consists of two steps: (i) *Self multilingual preference pair generation*: Multilingual prompts are used to generate multilingual responses from $M_t$, respectively. Then, $M_t$ is utilized to perform mutual translations between responses in dominant language (e.g., *en*) and non-dominant languages (e.g., *es, de, ru*). Finally, the inherent language imbalance in LLMs is leveraged to construct multilingual preference pairs. (ii) *Multilingual preference optimization*: Multilingual preference pairs are constructed by $M_t$ itself, which are used for training via a DPO+NLL objective, resulting in model $M_{t+1}$. The whole process is iteratively repeated, enhancing the model's multilingual abilities across all languages in each subsequent iteration, until optimization saturates.

In our experiments, we begin with Meta-Llama-3-8B-Instruct (Meta, 2024) as the seed model and perform two rounds of iteration. Results demonstrate that multilingual preference optimization not only significantly enhances the instruction-following abilities of non-dominant languages compared to the seed model but also improves the performance of the dominant language. This means that during training, the model is not constrained by the initial performance of the dominant language, which is crucial for iterative self-improvement within our approach. Although this effect will gradually saturate as the performance gap between languages narrows, it presents an intriguing opportunity to bootstrap the multilingual performance of LLMs across all languages without the need for human-authored datasets.

## 2    LANGUAGE IMBALANCE DRIVEN REWARDING

Our approach first assumes access to an instruction-following language model, and a set of multilingual training prompts. Starting from any instruction model that possesses basic multilingual generation capabilities, each iteration consists of two steps, (i) *Self multilingual preference pair generation* and (ii) *Multilingual preference optimization*, as shown in Figure 1. For the $t^{th}$ iteration,

we use the current model $M_t$, with the seed model denoted as $M_0$. Step (i) generates multilingual preference pairs data for DPO training in step (ii). After training, the updated model $M_{t+1}$ is utilized as the initial weight in the next iteration.

**Initialization** Given an instruction-tuned model $M_0$ and a set of parallel multilingual instruction prompts $\mathcal{X}$, where $\mathcal{X}$ includes both the dominant language and non-dominant languages participating in the self-improving process, the model is updated iteratively, resulting in a sequence of models $M_1, M_2, \ldots, M_T$.

**Self Multilingual Preference Pair Generation** The current model $M_t$ generates corresponding responses $y_i^l \sim M_t(x_i^l)$ for the instruction $x_i^l$ in any language $l$ supported by the model.

$$y_i^l \sim M_t(x_i^l) \quad \text{for all } x_i^l \in \mathcal{X} \tag{1}$$

Let $dl$ and $nl$ represent any dominant and non-dominant language supported by the model, respectively. After generating the corresponding responses, we utilize the *self-translation* capability of LLM to facilitate translation between dominant and non-dominant language responses, the *self-translation* prompt is given in Appendix H.4. Specifically, for the dominant language response $y_i^{dl}$, $M_t$ is utilized to translate it into any non-dominant language $nl$, resulting in $y_i^{dl \to nl}$. Similarly, we randomly select a response in any non-dominant language $nl$ for each prompt, denoted as $y_i^{nl}$, and translate it into the dominant language, resulting in $y_i^{nl \to dl}$.

Due to the inherent differences in the multilingual capabilities of the model itself, and translation does not alter language biases. The following preference ranking holds true for the dominant language $dl$ and any non-dominant language $nl$ supported by the model:

For the same instruction in dominant language $x_i^{dl}$,

$$y_i^{dl} \succ y_i^{nl \to dl} \tag{2}$$

For the same instruction in non-dominant language $x_i^{nl}$,

$$y_i^{dl \to nl} \succ y_i^{nl} \tag{3}$$

Thus, the preference ranking between the dominant language and non-dominant languages is utilized to construct multilingual preference pair dataset in all languages supported by the model.

$$\mathcal{D}_{dl} = \left\{ x_i^{dl}, y_i^{dl}, y_i^{nl \to dl} \right\}_{i=1}^N \tag{4}$$

$$\mathcal{D}_{nl} = \left\{ x_i^{nl}, y_i^{dl \to nl}, y_i^{nl} \right\}_{i=1}^N \tag{5}$$

The $\mathcal{D}_{dl}$ and $\mathcal{D}_{nl}$ are combined to form the multilingual preference pair dataset, which is denoted as $\mathcal{D} = \{\mathcal{D}_{dl}, \mathcal{D}_{nl}\}$.

**Multilingual Preference Optimization** Given a multilingual preference pair $\{x, y_{win}, y_{lose}\}$ from $\mathcal{D}$, a variant of DPO is employed to maximize the probability of the chosen output $y_{win}$ and minimize that of the undesirable output $y_{lose}$. Specifically, a negative log-likelihood (NLL) loss term for the chosen labels is incorporated into the vanilla DPO (Rafailov et al., 2024) formulation to improve alignment performance. The optimization objective is formulated as:

$$\mathcal{L}_{\mathcal{DPO}} = -\mathbb{E}_{(x,y_{win},y_{lose})\sim\mathcal{D}} \left[ \log \sigma \left( \beta \log \frac{M_\theta(y_{win}|x)}{M_t(y_{win}|x)} - \beta \log \frac{M_\theta(y_{lose}|x)}{M_t(y_{lose}|x)} \right) \right] \tag{6}$$

$$\mathcal{L}_{\mathcal{NLL}} = -\mathbb{E}_{(x,y_{win})\sim\mathcal{D}} \left[ \frac{\log M_\theta(y_{win}|x)}{|y_{win}|} \right] \tag{7}$$

Overall,

$$\mathcal{L} = \mathcal{L}_{\mathcal{DPO}} + \alpha \mathcal{L}_{\mathcal{NLL}} \tag{8}$$

Where $M_\theta(\cdot|x)$ is the policy model to be optimized, $M_t(\cdot|x)$ is the reference model kept unchanged during training. The parameters $\theta$ are initialized from model $M_t$, $\sigma$ is the sigmoid function. Note that the $\mathcal{L}_{\mathcal{NLL}}$ term is normalized by the response length, while DPO loss is not.

After the DPO training, our next model is updated as $M_{t+1} = M_\theta$, which will be utilized to construct new multilingual preference pairs data for the next iteration.

**Iterative Training**   Our overall procedure starts from an instruction-following model $M_0$ and instruction prompts $\mathcal{X}$, training a series of models $M_1, M_2, ..., M_T$. The models and corresponding training data used are defined as follows: (1) $M_0$: Base LLM; Instruction-following model. (2) $M_1$: Initialized with $M_0$, using $M_0$ and $\mathcal{X}$ to generate $\mathcal{D}_0$, then conduct multilingual preference optimization on the $\mathcal{D}_0$. (3) $M_2$: Initialized with $M_1$, using $M_1$ and $\mathcal{X}$ to generate $\mathcal{D}_1$, then conduct multilingual preference optimization on the $\mathcal{D}_1$.

## 3  DISCUSSION

The insight behind our proposed method is to leverage *the inherent differences in the multilingual capabilities of LLMs to provide rewards for DPO training.* Therefore, two key questions remain to be addressed:

### 3.1  DO LLMS EXHIBIT SIGNIFICANT DIFFERENCES IN MULTILINGUAL CAPABILITIES?

While the differences in multilingual capabilities have been evidenced by many prior works (Ranaldi & Pucci, 2023; Yuan et al., 2023; Zhao et al., 2024), we further validate the disparity in multilingual capabilities in Llama-3.

Table 1:  The average quality of responses across different languages for parallel multilingual instructions. Note that Llama-3-8B-Instruct is subject to the off-target issue in certain languages (*e.g., ja and ru*).

| **Model** | GPT-4 Score (0-10) | | | | | | |
|---|---|---|---|---|---|---|---|
| | en | es | fr | it | de | ja | ru |
| **Llama-3-8B-Instruct** | **9.60** | 8.34 | 6.43 | 6.66 | 4.69 | 0.76 | 2.21 |

Specifically, we randomly selected 100 multilingual Alpagasus (Chen et al., 2023a) instructions and evaluated the response quality across different languages using GPT-4 score. Based on the technical report for LLaMA 3 (Meta, 2024), English is selected as the dominant language, while the other languages are considered non-dominant languages. As shown in Table 1, a significant difference in response quality remains between the dominant language (*en*) and non-dominant languages, demonstrating an inherent imbalance exists in the multilingual capabilities within the model.

### 3.2  DOES TRANSLATION PRESERVE THE RANKING OF RESPONSE PREFERENCES?

As shown in Discussion 3.1, *Given an English prompt $x_i^{en}$ and a non-dominant language prompt $x_i^{nl}$, model M consistently produces a better response for the English prompt:* $M(y_i^{en}|x_i^{en}) \succ M(y_i^{nl}|x_i^{nl})$. However, self-translation is employed by our method to convert the English response $y_i^{en}$ into non-dominant language $nl$, and vice versa. Therefore, a key question arises: Do the translated responses preserve the preference ranking in Equation 2 and 3?

As translation will largely preserve the semantics and the structure of the sentence, it is reasonable to believe that the preference ranking stemming from the quality difference of the response is largely preserved. To verify our assumption, the GPT-4 score is utilized to assess the quality of self-translated responses and compare it with original responses. As shown in Table 2, the GPT-4 score of the translated response $y^{dl \rightarrow nl}$ is lower than the original response $y^{dl}$. However, a substantial gap remains between the translated response and the original response in non-dominant languages, which is consistent with the original preference ranking. This conclusion also holds for the dominant language (English), where the original response is superior to the response sampled from non-dominant languages and self-translated into English. Overall, the self-translation process does preserve the ranking of response preference.

To observe the final preference ranking, multilingual preference pairs are constructed between the original and translated responses, sampling 100 pairs from each language. Reward accuracy (Win Rate) was then assessed through head-to-head comparisons by GPT-4. As shown in Table 3, language imbalance provides a positive reward (>0.50). Moreover, the strength of reward signals varies across languages, ranging from 0.57 in *es* to 0.79 in *ja*. In line with the GPT-4 score in Table 1, this indicates that languages with weaker performance in LLMs tend to exhibit stronger reward signals.

Table 2: The average quality of self-translated responses. We selected the same responses discussed in Section 3.1 and assessed the self-translate quality using GPT-4 score. The underlined scores represent the self-translation of responses sampled from other languages into English ($y^{nl \to en}$), while the **bold** scores indicate the self-translation of English responses into other languages ($y^{en \to nl}$).

| Type | GPT-4 Score (0-10) | | | | | | |
|---|---|---|---|---|---|---|---|
| | en | es | fr | it | de | ja | ru |
| Self Generation | 9.60 | 8.34 | 6.43 | 6.66 | 4.69 | 0.76 | 2.21 |
| Self Translation | 8.03 | **9.32** | **9.17** | **8.72** | **8.96** | **7.89** | **7.75** |

Based on empirical observations, languages are classified with rewards of 0.60 as the threshold into low-reward ones *(es, fr, it)* and high-reward ones *(de, ja, ru)*.

Table 3: The reward accuracy of multilingual preference pairs.

| Model | Reward Accuracy (0-1) | | | | | | |
|---|---|---|---|---|---|---|---|
| | en | es | fr | it | de | ja | ru |
| Llama-3-8B-Instruct | 0.72 | 0.57 | 0.60 | 0.57 | 0.70 | 0.79 | 0.74 |

## 4 GENERAL INSTRUCTION FOLLOWING

### 4.1 EXPERIMENTAL SETUP

**Base Models**   In our experiments, we use a widely adopted instruction-following model as our base model $M_0$, namely Llama-3-8B-Instruct (Meta, 2024). Llama-3-8B-Instruct, as an English-centric LLM, often encounters off-target issues when handling non-English requests.

**Languages**   English is chosen as the common dominant language, and non-dominant languages include high-reward languages (German, Russian) and low-reward languages (Spanish, French). Additionally, Chinese is selected as an unseen language to observe the generalization of our approach. Note that unseen language means that not included in the training data.

**Datasets**   The Alpagasus dataset (Chen et al., 2023a) includes 9,000 high-quality instruction-following examples filtered from the original 52,000 in the Alpaca dataset (Taori et al., 2023). We sample 1,000 prompts from the Alpagasus dataset and translate them into other languages using the Google Translate API to obtain multilingual prompts.

**Implementation Details**   Models are trained for one epoch in each iteration across all experiments. More details are described in Appendix F.4.

**Evaluation and Metrics**

- **Head-to-head performance**: Head-to-head performance is evaluated between base model and the iterative models using GPT-4 Turbo as an evaluator (Liu et al., 2023) over 805 test prompts in X-AlpacaEval (Zhang et al., 2023). The detailed setup can be found in Appendix F.1.

- **X-AlpacaEval leaderboard**: We extend the existing AlpacaEval 2.0 toolkit (Li et al., 2023d) from an English-only framework to a multilingual one and compare both proprietary and open-source models on their multilingual instruction-following abilities.

- **Multilingual MT-Bench**: Results are additionally reported on multilingual MT-Bench. MT-Bench (Zheng et al., 2024a) consists of a series of open-ended questions that evaluate the multi-turn conversational and instruction-following abilities, which uses GPT-4 Turbo to grade the model responses on a scale of 10.

- **Multilingual NLP benchmarks**: To assess the *alignment tax* of our method, we further evaluate the performance of our model on multilingual versions of the MMLU (Hendrycks

et al., 2020), HellaSwag (Zellers et al., 2019), ARC Challenge (Clark et al., 2018) and TruthfulQA (Lin et al., 2021) benchmarks.

**Fair Evaluation** Appendix D.1 explains how to prevent language bias in LLM-as-a-Judge and D.2 highlights GPT-4's multilingual judging capabilities, aligning with the advanced GPT-4o. D.3 discusses avoiding translationese bias in evaluation.

## 4.2 HEAD-TO-HEAD PERFORMANCE

The head-to-head win rates of our models on the X-AlpacaEval dataset are shown in Figure 2.

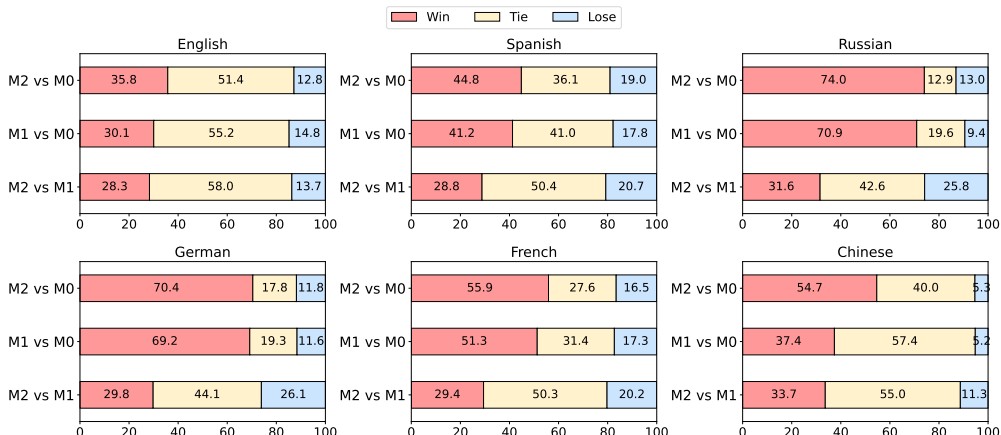

Figure 2: Multilingual Instruction following ability improves with *Language Imbalance Driven Rewarding* on Llama-3-8B-Instruct model.

**Findings 1: Language Imbalance Driven Rewarding is effective.** The head-to-head performance shows that $M_1$ achieves notable win rates against $M_0$ across all five training languages and one unseen language. For these five training languages, $M_1$ demonstrates a significant improvement, with $\Delta$**W-L** values for each language ranging from 15.3% (*en*) to 61.5% (*ru*) compared to the base model. Upon comparing different languages, high-reward languages (*ru, de*) gain larger improvements than low-reward languages (*es, fr*) in Iteration 1 ($M_1$ vs. $M_0$), but in Iteration 2 ($M_2$ vs. $M_1$), the gains across all languages diminish and converge. We hypothesize that the reward signals across different languages gradually weaken and align over iterations.

**Findings 2: The dominant language also benefits from Language Imbalance Driven Rewarding.** Since the responses for non-dominant languages are translated from English, it is natural for these languages to see improvements. However, English also achieves better performance compared to the reference model. As shown in Figure 2, English shows a 15.3% increase in $\Delta$**W-L** for $M_1$ vs. $M_0$, and a 14.6% increase in $\Delta$**W-L** for $M_2$ vs. $M_1$. These results indicate that the dominant language also benefits from rejected responses translated from non-dominant languages, highlighting the value of incorporating negative samples in preference pair construction, aligning with observation in Duan et al. (2024).

**Findings 3: Iterative training is possible and effective.** Findings 2 reveals that English benefits from language imbalance driven rewarding, which lays the foundation for iterative optimization. Specifically, the enhancement in English makes it possible to generate higher-quality responses in the next iteration of training, enabling continual self-improving. In Figure 2, a consistent gain is observed in Iteration 1 ($M_1$ vs. $M_0$) and Iteration 2 ($M_2$ vs. $M_1$) in all languages. Compared to Iteration 1, the gains for all training languages (except for English) in Iteration 2 become more consistent and convergent. This demonstrates that our approach are capable of iteratively aligning all languages until reaching saturation.

**Findings 4: Multilingual optimization can generalize to unseen languages.** For the unseen language, the gains ($\Delta$**W-L**) in Chinese are 32.2% for Iteration 1 ($M_1$ vs. $M_0$) and 22.4% for Iter-

ation 2 ($M_2$ vs. $M_1$), respectively. These results indicate that multilingual preference optimization facilitates cross-lingual transfer, which is consistent with observations in Dang et al. (2024).

Table 4: The X-AlpacaEval Leaderboard, which shows the win rate over GPT-4 Turbo evaluated by GPT-4.

| Model | Win Rate | | | | | Avg |
|---|---|---|---|---|---|---|
| | en | es | ru | de | fr | |
| *Language Imbalance Driven Rewarding* | | | | | | |
| Meta-Llama-3-8B-Instruct (M0) | 24.90% | 18.08% | 7.81% | 8.65% | 14.18% | 14.72% |
| Iteration 1 (M1) | 30.11% | 21.82% | 18.01% | 16.87% | 17.51% | 20.86% |
| Iteration 2 (M2) | 34.09% | 21.21% | 19.25% | 16.02% | 20.34% | 22.18% |
| *Multilingual Alignment* | | | | | | |
| Meta-Llama-3-8B-Instruct (SFT) | 21.88% | 18.98% | 15.90% | 16.68% | 18.15% | 18.32% |
| *SOTA Multilingual Models* | | | | | | |
| GPT-4o-mini | 45.17% | 44.63% | 47.03% | 44.2% | 44.93% | 45.19% |
| GPT-4-0613 | 15.61% | 18.18% | 16.82% | 16.00% | 15.23% | 16.37% |
| GPT-3.5-turbo-0125 | 11.96% | 14.42% | 13.74% | 12.41% | 12.70% | 13.05% |
| Qwen2-72B-Instruct | 37.72% | 24.73% | 27.15% | 23.93% | 24.63% | 27.63% |
| Meta-Llama-3-70B-Instruct | 39.74% | 32.58% | 9.14% | 9.48% | 25.20% | 23.23% |
| InternLM2.5-Chat-20B | 31.77% | 16.62% | 11.10% | 11.56% | 13.61% | 16.93% |
| Qwen1.5-14B-Instruct | 22.15% | 20.63% | 12.02% | 16.05% | 18.55% | 17.88% |
| Meta-Llama-2-13B-Instruct | 8.84% | 5.31% | 0.93% | 1.19% | 1.36% | 3.53% |
| PolyLM-Chat-13B | 3.81% | 3.61% | 2.27% | 2.79% | 3.56% | 3.21% |
| Aya-23-8B | 15.26% | 16.68% | 17.95% | 18.50% | 14.70% | 16.62% |
| Qwen2-7B-Instruct | 24.39% | 13.89% | 14.33% | 11.45% | 15.97% | 16.01% |
| Mistral-7B-Instruct-v0.3 | 21.46% | 13.36% | 13.75% | 11.91% | 13.28% | 14.75% |

## 4.3 X-ALPACAEVAL LEADERBOARD

The X-AlpacaEval leaderboard, as shown in Table 4, demonstrates a high degree of consistency with head-to-head evaluations. After two rounds of iteration, Llama-3-8B-Instruct achieves average improvements of 7.46% in win rates over GPT-4 Turbo across five languages. Additionally, we evaluate the performance of state-of-the-art multilingual models on X-AlpacaEval, including OpenAI's GPT-4o, GPT-4 (Achiam et al., 2023), along with Qwen series (Bai et al., 2023; Yang et al., 2024), Llama series (Touvron et al., 2023b; Meta, 2024), InternLM2 (Cai et al., 2024), Aya-23 (Üstün et al., 2024), Mistral (Jiang et al., 2023) and PolyLM (Wei et al., 2023). Our method based on Llama-3-8B-Instruct outperforms both 7B and 14B-level models, achieving comparable performance to the 70B-level models.

Moreover, a comparative experiment on *multilingual alignment* is conducted, which performed supervised fine-tuning by self-translating model responses from the dominant language to non-dominant languages under the same experimental conditions. Multilingual alignment utilizes the performance of the dominant language as an anchor to align the capabilities between dominant and non-dominant languages. While there is an improvement in performance on non-dominant languages, a significant gap remains compared to our method, with 18.32% vs. 22.18% in Llama3. Notably, multilingual alignment places excessive focus on non-dominant languages during the SFT process, resulting in a degradation of performance in English (-3.02%). In contrast, our approach improves English performance (+9.19%). This improvement is crucial for enabling iteration in our method, whereas the performance decline in English seen with multilingual alignment hinders further iteration.

## 4.4 PERFORMANCE ON MULTILINGUAL MT-BENCH

Table 5 reports the multilingual MT-Bench results on a scale of score 10. A significant performance improvement on MT-Bench in Llama3 is observed across the training iterations, increasing from

6.80 in $M_0$ to 7.51 in $M_2$. This is because Llama3 initially exhibits a strong reward signal in $M_0$; however, this signal weakens as iterations progress. A detailed analysis is provided in Section 4.6.

Table 5: The Multilingual MT-Bench Benchmark.

| Model | Training Languages | | | | | Unseen | Avg |
|---|---|---|---|---|---|---|---|
| | en | es | ru | de | fr | zh | |
| Meta-Llama-3-8B-Instruct (M0) | 8.20 | 7.51 | 5.86 | 6.36 | 7.21 | 5.64 | 6.80 |
| Iteration 1 (M1) | 8.22 | 7.55 | 7.12 | 7.46 | 7.56 | 5.87 | 7.30 |
| Iteration 2 (M2) | 8.30 | 7.59 | 7.37 | 7.62 | 7.93 | 6.22 | 7.51 |

## 4.5 ALIGNMENT TAX ON MULTILINGUAL NLP BENCHMARKS

Previous studies have shown that instruction tuning and RLHF can lead to forgetting, also known as the alignment tax (Ouyang et al., 2022). The changes in world knowledge and commonsense reasoning abilities are examined throughout the iterative process by evaluating its performance on multilingual NLP benchmarks.

Table 6 presents the average results across five training languages (English, Spanish, Russian, German, French) and one unseen language (Chinese) on four benchmarks, with more detailed results provided in the Appendix G.1. **Overall**, during the iteration process, the performance on the benchmarks not only exhibits no significant degradation compared to the base models but also shows a slight improvement. These results indicate that the multilingual preference optimization process did not introduce any alignment tax.

Table 6: The Multilingual NLP Benchmarks.

| Model | Multilingual MMLU | Multilingual HellaSwag | Multilingual ARC challenge | Multilingual TruthfulQA | |
|---|---|---|---|---|---|
| | | | | MC1 | MC2 |
| Meta-Llama-3-8B-Instruct (M0) | $0.5666_{\pm 0.0043}$ | $0.4724_{\pm 0.0051}$ | $0.4228_{\pm 0.0144}$ | $0.3417_{\pm 0.0168}$ | $0.5076_{\pm 0.0158}$ |
| Iteration 1 (M1) | $0.5687_{\pm 0.0043}$ | $0.4761_{\pm 0.0051}$ | $0.4312_{\pm 0.0144}$ | $0.3464_{\pm 0.0169}$ | $0.5169_{\pm 0.0157}$ |
| Iteration 2 (M2) | $0.5687_{\pm 0.0043}$ | $0.4763_{\pm 0.0051}$ | $0.4321_{\pm 0.0144}$ | $0.3472_{\pm 0.0169}$ | $0.5165_{\pm 0.0157}$ |

## 4.6 THE REWARD SIGNAL CHANGES OVER ITERATIONS, GETTING STRONGER OR WEAKER?

The reward signal strength on pairwise data was analyzed at the beginning of each iteration, as outlined in Section 3.2. Figure 3 shows the change in reward accuracy on training languages *(en, es, ru, de, fr)* and unseen languages *(it, ja)* across iterations. For the training languages, high-reward languages, except English, gradually shift to lower-reward status after Iteration 1. As English capabilities improve through iterations, low-reward languages remain in the lower-reward range with some fluctuations, enabling the self-improving process to continue iteratively.

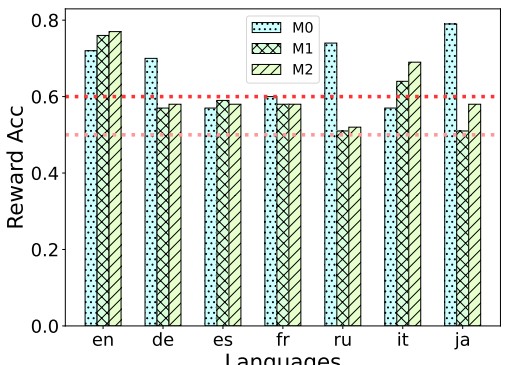

Figure 3: The reward accuracy over iterations.

For unseen languages, reward accuracy also steadily increases *(it)* as English capabilities improve continuously. However, during the DPO training, certain preferences, such as controlling off-target responses, are transferred to unseen languages *(ja)*. This is evident from the sharp drop in Japanese reward accuracy after Iteration 1, which corresponds to a reduction in off-target responses.

## 4.7 SCALING AND GENERALIZING

In Appendix C, we scale our method to Qwen2-7B-Instruct (Yang et al., 2024). In Appendix E, we extend our approach to extreme scenarios: using a weaker base model, Llama2-7B-Chat (Touvron

et al., 2023b) in E.1, addressing lower-resource languages (*bn, sw, th*) in E.2, and relaxing the self-improvement paradigm through the use of an external translation system in E.3.

## 5 ARITHMETIC REASONING

Arithmetic reasoning is a task where language models often struggle (Ahn et al., 2024), and while they are considered language-agnostic (Brannon, 2005), existing LLMs demonstrate inconsistent reasoning capabilities across different languages. We scale our method to arithmetic reasoning to enhance reasoning performance across languages.

### 5.1 EXPERIMENTAL SETUP

**Experiments Settings** The arithmetic reasoning task is conducted on Llama-3-8B-Instruct, starting with multilingual GSM8K (Cobbe et al., 2021) prompts. Performance was measured using the MGSM benchmark (Shi et al., 2022), which consists of 250 manually translated GSM8K problems in ten languages. We report reasoning accuracy (**Acc**) across five training and five unseen languages, and assess the off-target rate (**Off-tag**) of the reasoning responses using the LangDetect library. The implementation details are described in Appendix F.2.

**Compared Methods** We first compared our approach to *multilingual alignment*, where English responses were self-translated into other languages for SFT training. Additionally, we focus on comparing reasoning task performance with MAPO (She et al., 2024). To ensure a fair comparison with MAPO, we considered two variants: MAPO[†] uses the same sampling count as our method, while MAPO[‡] uses MAPO's sampling configuration but with training data size consistent with ours. We used MAPO's official code and hyperparameters for sampling and trained all preference pairs under identical training conditions. We report MAPO's best results, achieved after two iterations.

Table 7: Model performances on MGSM benchmark on LLama-3-8B-Instruct as base model. The subscript values represent the relative change in performance compared to the base model $M_0$ for each language. Improvements are indicated in green, and declines in red.

| Training Languages | en | | es | | ru | | de | | fr | |
|---|---|---|---|---|---|---|---|---|---|---|
| | Acc(↑) | Off-tag(↓) | Acc(↑) | Off-tag(↓) | Acc(↑) | Off-tag(↓) | Acc(↑) | Off-tag(↓) | Acc(↑) | Off-tag(↓) |
| M0 | 0.700 | 0 | 0.456 | 0.012 | 0.488 | 0.076 | 0.468 | 0.016 | 0.464 | 0.016 |
| *Multilingual Alignment* | 0.680 -2.0% | 0 | 0.604 +14.8% | 0 | 0.592 +10.4% | 0 | 0.552 +8.4% | 0.008 | 0.540 +7.6% | 0 |
| MAPO[†] | 0.668 -3.2% | 0 | 0.600 +14.4% | 0 | 0.608 +12.0% | 0.012 | 0.560 +9.2% | 0.028 | 0.524 +6.0% | 0.004 |
| MAPO[‡] | 0.716 +1.6% | 0 | 0.628 +17.2% | 0 | 0.620 +13.2% | 0.036 | 0.508 +4.0% | 0.028 | 0.592 +12.8% | 0.02 |
| M1 | 0.712 +1.2% | 0 | 0.616 +16.0% | 0 | 0.604 +11.6% | 0.004 | 0.564 +9.6% | 0 | 0.596 +13.2% | 0 |
| M2 | 0.720 +2.0% | 0 | 0.640 +18.4% | 0 | 0.620 +13.2% | 0.004 | 0.570 +10.2% | 0 | 0.608 +14.4% | 0 |

| Unseen Languages | ja | | sw | | th | | zh | | bn | |
|---|---|---|---|---|---|---|---|---|---|---|
| | Acc(↑) | Off-tag(↓) | Acc(↑) | Off-tag(↓) | Acc(↑) | Off-tag(↓) | Acc(↑) | Off-tag(↓) | Acc(↑) | Off-tag(↓) |
| M0 | 0.284 | 0.280 | 0.192 | 0.204 | 0.324 | 0.124 | 0.464 | 0.336 | 0.328 | 0.024 |
| *Multilingual Alignment* | 0.356 +7.2% | 0.020 | 0.216 +2.4% | 0.032 | 0.436 +11.2% | 0.004 | 0.520 +5.6% | 0.124 | 0.308 -2.0% | 0 |
| MAPO[†] | 0.348 +6.4% | 0.052 | 0.292 +10.0% | 0.044 | 0.508 +18.4% | 0.040 | 0.540 +7.6% | 0.172 | 0.324 -0.4% | 0 |
| MAPO[‡] | 0.384 +10.0% | 0.072 | 0.308 +11.6% | 0.016 | 0.472 +14.8% | 0.084 | 0.540 +7.6% | 0.104 | 0.384 +5.6% | 0.004 |
| M1 | 0.428 +14.4% | 0.008 | 0.320 +12.8% | 0.016 | 0.524 +20.0% | 0 | 0.552 +8.8% | 0.076 | 0.464 +13.6% | 0 |
| M2 | 0.476 +19.2% | 0.004 | 0.328 +13.6% | 0.002 | 0.536 +21.2% | 0 | 0.592 +12.8% | 0.064 | 0.472 +14.4% | 0 |

### 5.2 RESULTS

The results are presented in Table 7. In training languages, our approach outperforms multilingual alignment, with $M_2$ improving by 11.6% on average, compared to 7.8% for multilingual alignment. While multilingual alignment struggles to balance English and non-dominant languages, leading to a decline in English performance (-2.0%). In unseen languages, our method ($M_1$) achieves a 16.2% average improvement, exceeding the 11.6% gain observed in training languages. This suggests our approach effectively leverages language imbalance to learn language-agnostic reasoning, leading to superior generalization. In contrast, traditional multilingual alignment tends to overemphasize training languages to align English capabilities, resulting in much poorer generalization on unseen languages compared to our method (average 4.9% vs. 16.2%).

With the same sampling effort, MAPO[†] underperforms across all languages, likely due to its limited preference data. This highlights the efficiency of our method in directly leveraging language imbalance. Using the same training data, MAPO[‡] also performs worse than our approach in most languages, further demonstrating the superior effectiveness of our reward mechanism.

## 6    RELATED WORK

**LLM Self-Improving**    The goal of LLM self-improvement is to enhance its capability by leveraging the knowledge embedded within the model itself. Self-improvement can be broadly divided into two categories: *self-synthetic* and *self-critical*. *Self-synthetic* involves generating synthetic training data using the model itself. For example, Self-Instruct (Bai et al., 2022; Wang et al., 2022) is a technique for generating prompts and responses independently, which can be utilized to enhance a base language model. Instruction backtranslation (Li et al., 2023c) similarly augments and curates training data by augmenting it through back-translation from web documents to generate instructions. *Self-critical* (Dubois et al., 2024; Saha et al., 2023; Bai et al., 2024) refers to using LLM-as-a-Judge to assess the quality of the data. Self-rewarding (Yuan et al., 2024) involves using the model itself, via LLM-as-a-Judge prompting, to provide its own reward mechanism. However, existing self-improvement methods primarily focus on enhancing the overall capabilities of language models, without addressing the potential for self-improvement across different languages within the model. This is the key insight of our work.

**Multilingual LLMs**    Contemporary LLMs (Touvron et al., 2023a;b; Team et al., 2024; Bai et al., 2023; Achiam et al., 2023) are predominantly trained on multilingual corpora. However, the language distribution in the data primarily focuses on *English* and *Chinese*. The imbalanced data distribution above has led to significant limitations in the capabilities of LLMs across most languages. To enhance the multilingual capabilities, one straightforward approach is *multilingual training*, using multilingual data during the pre-training (Conneau & Lample, 2019; Le Scao et al., 2023), instruction-following (Li et al., 2023b; Muennighoff et al., 2022) and post training (Dang et al., 2024). However, high-quality multilingual data, particularly for low-resource languages, remains scarce and expensive. The second approach is *cross-lingual alignment*, which seeks to bridge the performance gap by aligning non-dominant and dominant languages. This approach utilizes techniques such as cross-lingual transfer (Etxaniz et al., 2023; Huang et al., 2023; Ranaldi & Pucci, 2023; Qin et al., 2023), cross-lingual instruction tuning (Schuster et al., 2019; Wen-Yi & Mimno, 2023) and self-distillation (Zhang et al., 2024).

The most similar work, MAPO (She et al., 2024), uses an off-the-shelf translation model as a reward model to assess cross-language consistency as the preference for optimization, focusing on aligning non-dominant languages with dominant ones in reasoning task. However, MAPO may struggle with consistency due to the limited context size in the translator, which makes it primarily suitable for reasoning tasks. In contrast, our approach relies on the LLM itself for translation, constructs preference pairs directly based on language imbalance, and supports both dominant and non-dominant languages. Our work enables iterative self-improvement across all languages for general tasks. The comparisons with MAPO on reasoning task are presented in Section 5.

## 7    CONCLUSION

This paper introduces *Language Imbalance Driven Rewarding*, which leverages the inherent imbalance between dominant and non-dominant languages in LLMs as a reward signal to bootstrap LLMs' multilingual capabilities in a self-improving manner. Starting from any instruction-following model with basic multilingual capabilities, this approach generates and self-translates the responses between dominant and non-dominant languages within LLMs, constructing preference ranking and adopting an Iterative DPO for training. This approach not only enhances LLM performance in non-dominant languages but also improves the dominant language's capacity. Experiments on Llama-3-8B-Instruct demonstrate significant improvements in instruction-following and arithmetic reasoning tasks. While much remains to be explored, this work paves the way for developing models capable of enhancing their multilingual abilities autonomously across all languages.

## ACKNOWLEDGEMENTS

This work is supported by the National Key R&D Program of China 2022ZD0160602. We would like to thank the anonymous reviewers for their helpful discussions and valuable comments.

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

APPENDIX

## A   LIMITATIONS AND FUTURE WORK

Our work has certain limitations. The reward signal is derived from the inherent language imbalance within LLMs, which provides a coarse-grained signal. Developing more refined and accurate reward signals for multilingual self-improvement is an area we plan to explore in future work. Additionally, our approach relies on the LLM to self-translate the multilingual responses. Although LLMs outperform traditional machine translation systems, the translated responses may still exhibit artifacts, which hinders the response quality.

## B   REPRODUCIBILITY STATEMENT

Codes and model weights have been made public after review to advocate future research. For evaluation, we primarily use greedy decoding to ensure reproducibility, except where specific generation configurations are mandated by certain benchmark tools. Note that evaluations on instruction-following abilities (AlpacaEval and MT-Bench) rely on OpenAI's API. The randomness of API responses may have little impact on the reproducibility of these benchmarks.

## C   SCALING ON MODEL: QWEN2

Qwen2-7B-Instruct (Yang et al., 2024) exhibits stronger multilingual capabilities and seldom produces off-target responses. We believe scaling our experiments to a multilingual LLM enhances the comprehensiveness of the evaluation. Following the experimental setup outlined in Section 4, Qwen2-7B-Instruct was chosen as the base model to validate the generalizability of *Language Imbalance Driven Rewarding*.

### C.1   HEAD-TO-HEAD PERFORMANCE

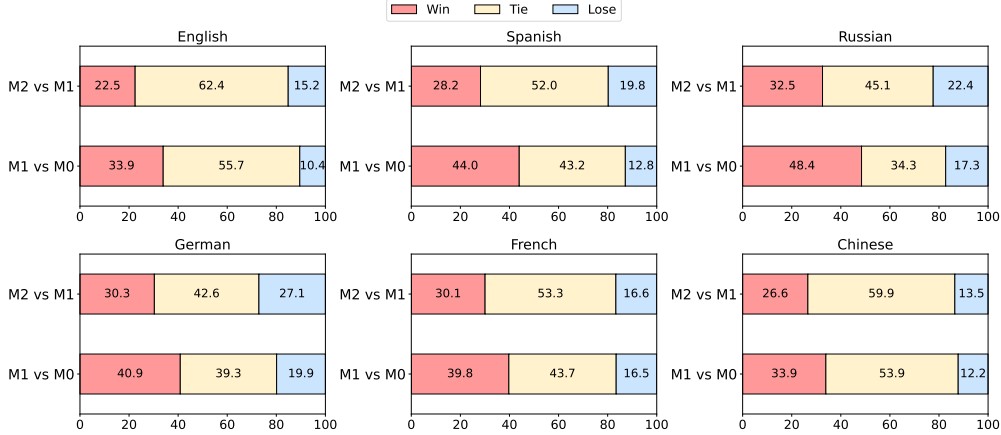

Figure 4: Multilingual Instruction following ability improves with *Language Imbalance Driven Rewarding* on Qwen2-7B-Instruct model.

Figure 4 illustrates Qwen2's head-to-head performance, which is highly consistent with the results from the Llama3 performance.

For the training languages, $M_1$ demonstrates a significant improvement, with $\Delta$**W-L** ranging from 21.0% to 31.2% compared to the base model. It demonstrates that Language Imbalance Driven Rewarding is effective. For the dominant language, English in $M_1$ gains 23.5% $\Delta$**W-L** compared with $M_0$, while English in $M_2$ gains 7.3% $\Delta$**W-L** compared with $M_1$. These results demonstrate that incorporating negative samples in preference pair construction significantly enhances the model's performance.

### C.2   X-ALPACAEVAL LEADERBOARD

The X-AlpacaEval leaderboard on Qwen2 as the base model is shown in Table 8. After two rounds of iterations, Qwen2-7B-Instrct achieved average improvements of 5.84% in win rates over GPT-4 Turbo across five languages, demonstrating performance comparable to 70B-level models.

Table 8: The X-AplacaEval Leaderboard On Qwen2-7B-Instruct, which shows the win rate over GPT-4 Turbo evaluated by GPT-4.

| Model | Win Rate | | | | | Avg |
| --- | --- | --- | --- | --- | --- | --- |
| | en | es | ru | de | fr | |
| *Language Imbalance Driven Rewarding* | | | | | | |
| Qwen2-7B-Instruct (M0) | 24.39% | 13.89% | 14.33% | 11.45% | 15.97% | 16.01% |
| Iteration 1 (M1) | 32.11% | 18.40% | 18.61% | 14.36% | 18.47% | 20.39% |
| Iteration 2 (M2) | 34.84% | 18.99% | 20.28% | 14.11% | 21.03% | 21.85% |
| *Multilingual Alignment* | | | | | | |
| Qwen2-7B-Instruct (SFT) | 20.79% | 18.27% | 19.36% | 12.61% | 17.39% | 17.68% |

Compared to multilingual alignment, our approach shows significantly better performance, with Win Rate of 21.85% versus 17.68% for multilingual alignment. We analyze that multilingual alignment overemphasizes non-dominant languages during supervised fine-tuning, resulting in a 3.6% decline in English performance. In contrast, our method effectively utilizes language imbalance as a reward signal, capturing the partial order relationships among all languages, including the dominant one. This strategy results in a significant improvement in English performance, with an increase of 10.54%.

## C.3 MULTILINGUAL MT-BENCH

In Table 9, Qwen2-7B-Instruct initially achieved a high score of 8.05, reflecting its robust multilingual capabilities. Despite the strong performance reducing the effectiveness of the language imbalance-driven reward signal, Qwen2 improved its average score to 8.20 after two training iterations. This shows that even with high initial scores, our approach continues to improve performance through iterative refinement.

Table 9: The Multilingual MT-Bench Benchmark On Qwen2-7B-Instruct.

| Model | Training Languages | | | | | Unseen | Avg |
| --- | --- | --- | --- | --- | --- | --- | --- |
| | en | es | ru | de | fr | zh | |
| Qwen2-7B-Instruct (M0) | 8.35 | 7.87 | 7.81 | 7.99 | 7.92 | 8.39 | 8.05 |
| Iteration 1 (M1) | 8.39 | 8.00 | 7.90 | 8.03 | 7.99 | 8.42 | 8.12 |
| Iteration 2 (M2) | 8.46 | 8.10 | 7.94 | 8.00 | 8.19 | 8.54 | 8.20 |

## C.4 MULTILINGUAL NLP BENCHMARKS

Table 10 shows average performance across five training languages and Chinese on four benchmarks based on Qwen2, detailed in Appendix G.1. Slight performance improvements are observed in multilingual optimization iterations compared to the base models, indicating that the multilingual alignment process does not incur any alignment tax.

Table 10: The Multilingual NLP Benchmark On Qwen2-7B-Instruct.

| Model | Multilingual MMLU | Multilingual HellaSwag | Multilingual ARC challenge | Multilingual TruthfulQA MC1 | MC2 |
| --- | --- | --- | --- | --- | --- |
| Qwen2-7B-Instruct (M0) | $0.6387_{\pm 0.0041}$ | $0.5139_{\pm 0.0052}$ | $0.4321_{\pm 0.0144}$ | $0.3731_{\pm 0.0172}$ | $0.5395_{\pm 0.0160}$ |
| Iteration 1 (M1) | $0.6402_{\pm 0.0041}$ | $0.5143_{\pm 0.0052}$ | $0.4316_{\pm 0.0144}$ | $0.3744_{\pm 0.0172}$ | $0.5418_{\pm 0.0159}$ |
| Iteration 2 (M2) | $0.6403_{\pm 0.0041}$ | $0.5130_{\pm 0.0052}$ | $0.4338_{\pm 0.0144}$ | $0.3740_{\pm 0.0172}$ | $0.5410_{\pm 0.0159}$ |

# D  DISCUSSION ON FAIR EVALUATION

## D.1  HOW TO AVOID LANGUAGE BIAS IN LLM-AS-A-JUDGE

The prior work (Hada et al., 2023) discussed that GPT-4's scores across languages may not be entirely consistent, as its evaluation capabilities can vary depending on the languages. In Table 1, we evaluate the GPT score for individual responses in different languages, e.g., $y^{en}$ or $y^{ru}$. This approach may introduce language bias, as scores could vary depending on the language in which the response is generated. To mitigate this potential issue, we perform more detailed pairwise comparisons within the same language in Tables 2 and 3, thereby avoiding cross-lingual scoring bias. Importantly, this language bias in Table 1 does not affect our conclusion, as we focus on constructing pairwise comparisons between responses within the same language for Tables 2 and 3.

Specifically, in each column of Table 2, both responses are in the same language. Take *ru* as an example. The *'Self Generation'* GPT score is calculated on $y^{ru}$ and the *'Self Translation'* GPT score is calculated on $y^{en \to ru}$. As both responses are in *ru*, it will not introduce potential language bias of LLM-as-a-Judge, providing a fair comparison. Similarly, in Table 3, the reward accuracy is evaluated on preference pairs consisting of $y^{ru}$ and $y^{en \to ru}$, which also avoids the cross-lingual comparison issue.

Both Table 2 and 3 evaluate GPT score based on pairwise comparisons within the same language. This methodology inherently avoids the cross-lingual comparison issue, ensuring a fairer and more consistent assessment since all evaluations are conducted within the same language.

## D.2  ALIGNING WITH ADVANCED MODEL: USING GPT-4O AS A JUDGE

To investigate this potential bias, we employed the more advanced GPT-4o model (`gpt-4o-2024-08-06`) for evaluation. Specifically, we used the same responses as those in Table 1 and evaluated their quality using GPT-4o. The results in Table 11 highlight an inherent imbalance in the model's multilingual capabilities, which aligns with the findings from GPT-4 (`gpt-4-1106-preview`) evaluations in Table 1. The consistent results from GPT-4 (Table 1) and GPT-4o (Table 11) across different evaluation models indicate the robustness of our findings, even in the presence of potential cross-linguistic evaluation biases.

Table 11: The average quality of responses across different languages for parallel multilingual instructions, evaluated with **GPT-4o**.

| Model | GPT-4o Score (0-10) | | | | | | |
|---|---|---|---|---|---|---|---|
| | en | es | fr | it | de | ja | ru |
| Llama-3-8B-Instruct | **9.68** | 8.99 | 7.64 | 7.65 | 6.40 | 2.97 | 4.51 |

Moreover, we evaluated win rates on the X-AlpacaEval benchmark using GPT-4o. It is worth noting that the existing AlpacaEval repository does not offer a GPT-4o evaluator configuration with human-alignment calibration. As a result, we had to use GPT-4o with a configuration calibrated for GPT-4 Turbo, which may introduce some bias.

The results in Table 12 remain consistent with Table 4, evaluated by GPT-4 Turbo. These findings demonstrate consistent multilingual performance improvements and validate the robustness of our approach, despite potential minor evaluation bias.

Table 12: The X-AplacaEval Leaderboard On **Meta-Llama-3-8B-Instruct**, evaluated with **GPT-4o**.

| Model | Win Rate | | | | | Avg |
|---|---|---|---|---|---|---|
| | en | es | ru | de | fr | |
| M0 | 35.12% | 26.60% | 8.35% | 9.93% | 19.21% | 19.84% |
| M1 | 38.44% | 34.28% | 25.79% | 26.34% | 30.13% | 31.00% |
| M2 | 39.78% | 33.68% | 28.37% | 26.67% | 31.49% | 32.00% |

By adopting these measures, we believe we have adequately addressed the uncertainty surrounding GPT-4's reliability as a multilingual evaluator, ensuring that the assessments are as fair and consistent as possible.

### D.3 How to avoid Translationese bias in multilingual benchmarks evaluation

Due to the expense and scarcity of multilingual benchmarks, most benchmarks in multilingual-related work, including both open-ended and structured tests, are predominantly machine-translated from English into other languages. Since the preference data is also constructed using translation, there is a possibility that "translationese bias" could be exploited. However, our approach leverages LLMs for self-translation to construct training data, which offers key advantages to avoid translationese bias:

(1) Different Data Distributions: Our method uses LLM self-translation to construct training data, while multilingual benchmarks are derived from machine translation of English datasets. This ensures that the training data and benchmark data have different distributions, effectively minimizing the risk of translationese bias influencing evaluation.

(2) Reduction of Translationese Artifacts: LLM self-translation significantly reduces translationese effects, producing fluent and natural translations that align closely with native text. This is supported by prior works (Chen et al., 2023c; Kunilovskaya et al., 2024), which highlights the high-quality outputs of LLMs.

## E Generalizing to extreme scenarios

### E.1 Performance on Weaker Model: Llama2

Table 13 demonstrates that even when starting with a model with weaker multilingual capabilities, such as Llama2-7B-Chat, which exhibits extremely low performance in languages like Russian (ru), German (de), and French (fr) on the X-AlpacaEval, significant improvements can be achieved. By leveraging language imbalance-driven rewarding for self-multilingual optimization across two iterations, the model shows substantial enhancement across all training languages, particularly in those where the original model's performance was initially weaker.

Table 13: The X-AplacaEval Leaderboard On **Llama-2-7B-Chat**, evaluated with **GPT-4o**.

| Model | Win Rate | | | | | Avg |
|-------|----------|------|------|------|------|------|
| | en | es | ru | de | fr | |
| M0 | 11.60% | 3.40% | 0.32% | 0.87% | 0.69% | 3.38% |
| M1 | 14.62% | 5.30% | 1.91% | 1.89% | 2.68% | 5.28% |
| M2 | 14.86% | 6.62% | 4.51% | 3.62% | 6.35% | 7.19% |

### E.2 Performance on Lower-resource languages: *bn, sw, th*

Llama3-8b-Instruct demonstrates weak performance in low-resource languages such as Bengali (bn), Swahili (sw), and Thai (th). It is important to note that the effectiveness of post-training on these low-resource languages is inherently limited. The model's multilingual capabilities are primarily developed during the pre-training phase, where it learns from a diverse and extensive multilingual corpus. As such, the gains from post-training are incremental and cannot fully overcome the limitations of the pre-training data for low-resource languages.

To assess the impact of our approach on these languages, we conducted experiments using Llama3-8b-Instruct as the base model. Table 14 shows that even though the model performs weakly in these languages, our approach remains effective in low-resource settings and can iteratively improve the model's performance across all languages.

### E.3 Relaxation of the Self-improvement paradigm under extreme scenarios

Our approach is designed as a self-improving paradigm, where the model iteratively refines its capabilities. The primary goal of using self-translation is to preserve the integrity of a self-improving paradigm. However, in cases where the model's generation capabilities are particularly limited for certain low-resource languages, relaxing this constraint and using an external translator is also a viable solution.

Table 14: The X-AplacaEval Leaderboard On **Meta-Llama-3-8B-Instruct** in **lower-resource languages with self-translation**, evaluated with **GPT-4o**.

| Model | Win Rate | | | | Avg |
|-------|----------|-----|-----|-----|-----|
|       | en | bn | sw | th | |
| M0 | 35.12% | 1.05% | 1.01% | 2.79% | 9.99% |
| M1 | 38.22% | 4.06% | 1.15% | 23.27% | 16.68% |
| M2 | 39.27% | 4.49% | 2.07% | 28.07% | 18.48% |

Table 15 demonstrates Google Translate as external translation systems can be leveraged for mutual translation between dominant and low-resource languages to bootstrap performance.

Compared with self-translation in Table 14, the external Google translation system provides higher-quality data for low-resource languages, enhancing the model's capabilities in these languages during optimization due to the model's initially weaker generation capabilities in these languages. However, Self-translation more effectively improves the performance of English because it avoids introducing external translations, maintaining a consistent generation space. This prevents disruption to English's established capabilities, leading to better performance improvements.

Table 15: The X-AplacaEval Leaderboard On **Meta-Llama-3-8B-Instruct** in **lower-resource languages with Google Translate System**, evaluated with **GPT-4o**.

| Model | Win Rate | | | | Avg |
|-------|----------|-----|-----|-----|-----|
|       | en | bn | sw | th | |
| M0 | 35.12% | 1.05% | 1.01% | 2.79% | 9.99% |
| M1 | 38.35% | 4.32% | 2.98% | 26.94% | 18.15% |
| M2 | 38.38% | 5.99% | 3.45% | 29.10% | 19.23% |

# F  IMPLEMENTATION DETAILS

## F.1  EXPERIMENTAL DETAILS ON GENERAL INSTRUCTION-FOLLOWING

**Head-to-head Performance**    Considering the excellent multilingual understanding ability of GPT-4, we use GPT-4[1] as a judge to conduct the automatic evaluation. GPT-4 as an evaluator has a higher correlation with human judgements (Liu et al., 2023; Li et al., 2023d).

Specifically, we use pairwise evaluation, asking GPT-4 to determine the better response between $(r_1, r_2)$ from different models, given instruction $x_i$. During the evaluation, GPT-4 assigns a score from 0 to 10 based on the prompt in Appendix H.2.

GPT-4 as an evaluator, exhibits a significant positional bias, showing a preference for responses that appear earlier (Zheng et al., 2024a). To mitigate this bias, we first request GPT-4 to evaluate $(r_1, r_2)$, then switch the position to $(r_2, r_1)$ for the second evaluation. The better response is the one that wins twice or wins once and draws once.

**X-AlpacaEval**    The X-AlpacaEval leaderboard lists the win rates of various models over GPT-4 Turbo evaluated against GPT-4. Based on the `weighted_alpaca_eval_gpt4_turbo` config used in AlpacaEval 2, we modified the prompt to enable the model to better evaluate multilingual responses. The modified prompt is provided in the Appendix H.3.

**Multilingual MT-Bench**    MT-Bench (Zheng et al., 2024a) is a challenging multi-turn English question set designed to evaluate the conversational and instruction-following ability of LLMs. In our experimental setup, we collect multilingual MT-Bench in German, French, Russian, and Chinese from Github[2]. In addition, we translate the English data into Spanish by Google Translate API.

---

[1] We use "gpt-4-1106-preview" API during the head-to-head evaluation.

[2] https://github.com/lightblue-tech/multilingual-mt-bench

**Multilingual NLP Benchmarks** We examine the changes in world knowledge and commonsense reasoning abilities throughout the iterative process by evaluating it on the multilingual versions of the MMLU (Hendrycks et al., 2020)[3], HellaSwag (Zellers et al., 2019)[4], ARC Challenge (Clark et al., 2018)[5] and TruthfulQA (Lin et al., 2021)[6] benchmarks. We utilized the multilingual benchmarks provided by Okapi (Lai et al., 2023), which were translated from the original benchmarks using ChatGPT, and conducted evaluations under the `lm-evaluation-harness` (Gao et al., 2024) framework.

## F.2 Experimental Details on Arithmetic Reasoning

**Datasets** We start from the GSM8K (Cobbe et al., 2021) dataset, which consists of 8.5K high-quality grade school math problems created by human problem writers in English. We utilize the instructions from the 7,473 training examples and translate them into multiple languages using the Google Translate API to construct the multilingual GSM8K instructions.

We input the multilingual GSM8K instructions into the model and explicitly constrain the model's response language in the prompt, as detailed in Appendix H.5, for both training and inference. We believe that by providing instructions in an explicit language and requiring the model to respond in that language, we can fully capture the model's reasoning abilities in that language. After obtaining the multilingual reasoning responses, we filter the responses with correct reasoning in English, followed by applying *Language Imbalance Driven Rewarding*.

## F.3 Experiments Environments

All experiments were conducted on Ubuntu 22.04 equipped with 8 NVIDIA A100 GPUs. Our code mainly depends on Python 3.10 and PyTorch 2.3.0. we fine-tune all models using LLaMA-Factory (Zheng et al., 2024b) framework, and inference models with vLLM (Kwon et al., 2023) framework. Training for all models was launched with the accelerate (Gugger et al., 2022) in DeepSpeed ZeRO Stage2 (Rajbhandari et al., 2021) and Flash-Attention 2 (Dao, 2023) mechanism.

## F.4 Hyperparameters

All models are optimized using AdamW (Kingma & Ba, 2014), with a cosine learning rate scheduler that includes a warm-up phase constituting 3% of the total training duration. DPO+NLL runs are trained with KL-penalty $\beta = 0.1$. The coefficient $\alpha$ is set to $1$ for all experiments in the paper. The details of hyperparameters are shown in Table 16.

Table 16: The hyperparameters on various experiments. 'LR' refers to the Learning Rate, and 'BS' denotes the Batch Size

| Experiments | LR | BS | Epoch |
|---|---|---|---|
| *Language Imbalance Driven Rewarding* | | | |
| General Instruction-following Task | 5e-7 | 16 | 1 |
| Arithmetic reasoning Task | 5e-6 | 64 | 1 |
| *Multilingual Alignment* | | | |
| All Tasks | 1e-5 | 128 | 3 |

## G Detailed Results and Analysis

### G.1 Multilingual NLP Benchmarks

We list the detailed information of the benchmarks as follows:

- MMLU (Massive Multitask Language Understanding) (Hendrycks et al., 2020) is a benchmark designed to evaluate the knowledge acquired during pre-training, focusing on zero-shot and few-shot settings. This makes it more challenging and closer to how humans are

---

[3]`https://huggingface.co/datasets/alexandrainst/m_mmlu`
[4]`https://huggingface.co/datasets/alexandrainst/m_hellaswag`
[5]`https://huggingface.co/datasets/alexandrainst/m_arc`
[6]`https://huggingface.co/datasets/alexandrainst/m_truthfulqa`

evaluated. The benchmark spans 57 subjects, including STEM, the humanities, and the social sciences. We test it in a 5-shot setting.

- HellaSwag (Zellers et al., 2019) is a challenging dataset for evaluating commonsense NLI, which is particularly difficult for state-of-the-art models but trivial for humans. We test it in a zero-shot setting.

- The AI2 Reasoning Challenge (ARC) dataset (Clark et al., 2018) is a multiple-choice question-answering dataset based on science exams for grades 3 to 9. It is divided into two partitions: Easy and Challenge, with the latter containing more difficult questions requiring reasoning. We test the ARC Challenge in a zero-shot setting.

- TruthfulQA (Lin et al., 2021) is a benchmark designed to evaluate whether a language model generates truthful answers. It consists of 817 questions across 38 categories, including health, law, finance, and politics. Since evaluating generation tasks for truthfulness is challenging, the benchmark provides two multiple-choice formats: MC1 (Single-true) and MC2 (Multi-true), testing the ability to identify true statements. We test it in a zero-shot setting.

We report the detailed results in Table 17 of multilingual NLP benchmarks. Although the model contains only 1,000 Alpagasus instructions for each language, we find that the model still shows slight improvements on these benchmarks during the iterative process. The results across multiple benchmarks indicate that our method does not introduce alignment tax.

Table 17: The Multilingual NLP Benchmarks.

| Model | Training Languages | | | | | Unseen | Avg |
|---|---|---|---|---|---|---|---|
| | en | es | ru | de | fr | zh | |
| *Multilingual MMLU, 5-shot* | | | | | | | |
| Meta-Llama-3-8B-Instruct (M0) | $0.6567_{\pm 0.0038}$ | $0.5771_{\pm 0.0043}$ | $0.5335_{\pm 0.0044}$ | $0.5506_{\pm 0.0043}$ | $0.5654_{\pm 0.0043}$ | $0.5162_{\pm 0.0044}$ | $0.5666_{\pm 0.0043}$ |
| Iteration 1 (M1) | $0.6585_{\pm 0.0038}$ | $0.5765_{\pm 0.0043}$ | $0.5380_{\pm 0.0044}$ | $0.5536_{\pm 0.0043}$ | $0.5678_{\pm 0.0043}$ | $0.5179_{\pm 0.0044}$ | $0.5687_{\pm 0.0043}$ |
| Iteration 2 (M2) | $0.6590_{\pm 0.0038}$ | $0.5778_{\pm 0.0043}$ | $0.5368_{\pm 0.0044}$ | $0.5529_{\pm 0.0043}$ | $0.5678_{\pm 0.0043}$ | $0.5178_{\pm 0.0044}$ | $0.5687_{\pm 0.0043}$ |
| Qwen2-7B-Instruct (M0) | $0.7062_{\pm 0.0037}$ | $0.6324_{\pm 0.0042}$ | $0.6095_{\pm 0.0043}$ | $0.6048_{\pm 0.0042}$ | $0.6348_{\pm 0.0042}$ | $0.6446_{\pm 0.0042}$ | $0.6387_{\pm 0.0041}$ |
| Iteration 1 (M1) | $0.7073_{\pm 0.0037}$ | $0.6333_{\pm 0.0042}$ | $0.6118_{\pm 0.0043}$ | $0.6068_{\pm 0.0042}$ | $0.6361_{\pm 0.0042}$ | $0.6460_{\pm 0.0042}$ | $0.6402_{\pm 0.0041}$ |
| Iteration 2 (M2) | $0.7055_{\pm 0.0037}$ | $0.6342_{\pm 0.0042}$ | $0.6117_{\pm 0.0043}$ | $0.6097_{\pm 0.0042}$ | $0.6346_{\pm 0.0042}$ | $0.6463_{\pm 0.0042}$ | $0.6403_{\pm 0.0041}$ |
| *Multilingual HellaSwag, 0-shot* | | | | | | | |
| Meta-Llama-3-8B-Instruct (M0) | $0.5764_{\pm 0.0049}$ | $0.4877_{\pm 0.0052}$ | $0.4326_{\pm 0.0051}$ | $0.4483_{\pm 0.0051}$ | $0.4715_{\pm 0.0052}$ | $0.4181_{\pm 0.0051}$ | $0.4724_{\pm 0.0051}$ |
| Iteration 1 (M1) | $0.5777_{\pm 0.0049}$ | $0.4919_{\pm 0.0052}$ | $0.4372_{\pm 0.0052}$ | $0.4511_{\pm 0.0051}$ | $0.4782_{\pm 0.0052}$ | $0.4204_{\pm 0.0051}$ | $0.4761_{\pm 0.0051}$ |
| Iteration 2 (M2) | $0.5791_{\pm 0.0049}$ | $0.4921_{\pm 0.0052}$ | $0.4376_{\pm 0.0052}$ | $0.4512_{\pm 0.0051}$ | $0.4768_{\pm 0.0052}$ | $0.4209_{\pm 0.0051}$ | $0.4763_{\pm 0.0051}$ |
| Qwen2-7B-Instruct (M0) | $0.6116_{\pm 0.0049}$ | $0.5272_{\pm 0.0052}$ | $0.4730_{\pm 0.0052}$ | $0.4659_{\pm 0.0052}$ | $0.5124_{\pm 0.0052}$ | $0.4932_{\pm 0.0052}$ | $0.5139_{\pm 0.0052}$ |
| Iteration 1 (M1) | $0.6124_{\pm 0.0049}$ | $0.5261_{\pm 0.0052}$ | $0.4753_{\pm 0.0052}$ | $0.4654_{\pm 0.0052}$ | $0.5129_{\pm 0.0052}$ | $0.4936_{\pm 0.0052}$ | $0.5143_{\pm 0.0052}$ |
| Iteration 2 (M2) | $0.6110_{\pm 0.0049}$ | $0.5255_{\pm 0.0052}$ | $0.4734_{\pm 0.0052}$ | $0.4640_{\pm 0.0052}$ | $0.5114_{\pm 0.0052}$ | $0.4929_{\pm 0.0052}$ | $0.5130_{\pm 0.0052}$ |
| *Multilingual ARC challenge, 0-shot* | | | | | | | |
| Meta-Llama-3-8B-Instruct (M0) | $0.5316_{\pm 0.0146}$ | $0.4162_{\pm 0.0144}$ | $0.3781_{\pm 0.0142}$ | $0.3978_{\pm 0.0143}$ | $0.4371_{\pm 0.0145}$ | $0.3761_{\pm 0.0142}$ | $0.4228_{\pm 0.0144}$ |
| Iteration 1 (M1) | $0.5324_{\pm 0.0146}$ | $0.4265_{\pm 0.0145}$ | $0.3867_{\pm 0.0142}$ | $0.4140_{\pm 0.0144}$ | $0.4465_{\pm 0.0145}$ | $0.3812_{\pm 0.0142}$ | $0.4312_{\pm 0.0144}$ |
| Iteration 2 (M2) | $0.5358_{\pm 0.0146}$ | $0.4299_{\pm 0.0145}$ | $0.3884_{\pm 0.0143}$ | $0.4183_{\pm 0.0144}$ | $0.4423_{\pm 0.0145}$ | $0.3778_{\pm 0.0142}$ | $0.4321_{\pm 0.0144}$ |
| Qwen2-7B-Instruct (M0) | $0.5102_{\pm 0.0146}$ | $0.4111_{\pm 0.0144}$ | $0.4098_{\pm 0.0144}$ | $0.3618_{\pm 0.0141}$ | $0.4277_{\pm 0.0145}$ | $0.4667_{\pm 0.0146}$ | $0.4312_{\pm 0.0144}$ |
| Iteration 1 (M1) | $0.5085_{\pm 0.0146}$ | $0.4145_{\pm 0.0144}$ | $0.4072_{\pm 0.0144}$ | $0.3678_{\pm 0.0141}$ | $0.4226_{\pm 0.0145}$ | $0.4692_{\pm 0.0146}$ | $0.4316_{\pm 0.0144}$ |
| Iteration 2 (M2) | $0.5128_{\pm 0.0146}$ | $0.4205_{\pm 0.0144}$ | $0.4038_{\pm 0.0144}$ | $0.3704_{\pm 0.0141}$ | $0.4311_{\pm 0.0145}$ | $0.4641_{\pm 0.0146}$ | $0.4338_{\pm 0.0144}$ |
| *Multilingual TruthfulQA MC1, 0-shot* | | | | | | | |
| Meta-Llama-3-8B-Instruct (M0) | $0.3611_{\pm 0.0168}$ | $0.3333_{\pm 0.0168}$ | $0.3541_{\pm 0.0170}$ | $0.3173_{\pm 0.0166}$ | $0.3355_{\pm 0.0168}$ | $0.3490_{\pm 0.0170}$ | $0.3417_{\pm 0.0168}$ |
| Iteration 1 (M1) | $0.3611_{\pm 0.0168}$ | $0.3257_{\pm 0.0167}$ | $0.3617_{\pm 0.0171}$ | $0.3135_{\pm 0.0165}$ | $0.3532_{\pm 0.0170}$ | $0.3629_{\pm 0.0171}$ | $0.3464_{\pm 0.0169}$ |
| Iteration 2 (M2) | $0.3599_{\pm 0.0168}$ | $0.3321_{\pm 0.0168}$ | $0.3604_{\pm 0.0171}$ | $0.3160_{\pm 0.0166}$ | $0.3532_{\pm 0.0170}$ | $0.3617_{\pm 0.0171}$ | $0.3472_{\pm 0.0169}$ |
| Qwen2-7B-Instruct (M0) | $0.4064_{\pm 0.0172}$ | $0.3676_{\pm 0.0172}$ | $0.3756_{\pm 0.0173}$ | $0.3439_{\pm 0.0169}$ | $0.3787_{\pm 0.0173}$ | $0.3668_{\pm 0.0172}$ | $0.3731_{\pm 0.0172}$ |
| Iteration 1 (M1) | $0.4064_{\pm 0.0172}$ | $0.3688_{\pm 0.0172}$ | $0.3756_{\pm 0.0173}$ | $0.3414_{\pm 0.0169}$ | $0.3825_{\pm 0.0173}$ | $0.3718_{\pm 0.0172}$ | $0.3744_{\pm 0.0172}$ |
| Iteration 2 (M2) | $0.4076_{\pm 0.0172}$ | $0.3663_{\pm 0.0172}$ | $0.3731_{\pm 0.0172}$ | $0.3376_{\pm 0.0169}$ | $0.3863_{\pm 0.0174}$ | $0.3731_{\pm 0.0172}$ | $0.3740_{\pm 0.0172}$ |
| *Multilingual TruthfulQA MC2, 0-shot* | | | | | | | |
| Meta-Llama-3-8B-Instruct (M0) | $0.5171_{\pm 0.0152}$ | $0.4989_{\pm 0.0157}$ | $0.5256_{\pm 0.0162}$ | $0.4890_{\pm 0.0157}$ | $0.5033_{\pm 0.0158}$ | $0.5119_{\pm 0.0163}$ | $0.5076_{\pm 0.0158}$ |
| Iteration 1 (M1) | $0.5142_{\pm 0.0152}$ | $0.5059_{\pm 0.0156}$ | $0.5328_{\pm 0.0161}$ | $0.5003_{\pm 0.0155}$ | $0.5233_{\pm 0.0156}$ | $0.5250_{\pm 0.0163}$ | $0.5169_{\pm 0.0157}$ |
| Iteration 2 (M2) | $0.5187_{\pm 0.0151}$ | $0.5051_{\pm 0.0156}$ | $0.5331_{\pm 0.0161}$ | $0.4986_{\pm 0.0155}$ | $0.5171_{\pm 0.0157}$ | $0.5266_{\pm 0.0163}$ | $0.5165_{\pm 0.0157}$ |
| Qwen2-7B-Instruct (M0) | $0.5733_{\pm 0.0154}$ | $0.5231_{\pm 0.0161}$ | $0.5319_{\pm 0.0163}$ | $0.5356_{\pm 0.0161}$ | $0.5451_{\pm 0.0159}$ | $0.5282_{\pm 0.0161}$ | $0.5395_{\pm 0.0160}$ |
| Iteration 1 (M1) | $0.5757_{\pm 0.0154}$ | $0.5239_{\pm 0.0160}$ | $0.5347_{\pm 0.0162}$ | $0.5325_{\pm 0.0161}$ | $0.5495_{\pm 0.0158}$ | $0.5344_{\pm 0.0160}$ | $0.5418_{\pm 0.0159}$ |
| Iteration 2 (M2) | $0.5726_{\pm 0.0154}$ | $0.5235_{\pm 0.0158}$ | $0.5346_{\pm 0.0162}$ | $0.5306_{\pm 0.0160}$ | $0.5515_{\pm 0.0157}$ | $0.5334_{\pm 0.0160}$ | $0.5410_{\pm 0.0159}$ |

# H  PROMPTS TEMPLATE

## H.1  GPT-4 SCORE PROMPT

---

**Prompt in GPT-4 Score**

You are a helpful assistant tasked with scoring answers for a given instruction in `[LANG]`.
Please evaluate the following answer based on the provided instruction in `[LANG]`. A good answer should adhere to these criteria:

1. It should be in `[LANG]`, unless the instruction explicitly requests a different language.
2. It should address the request made in the instruction.
3. It should be factually and semantically coherent.
4. It should be grammatically correct and fluent.
5. It should be helpful, relevant, detailed, and accurate.

```
<instruction>
[INSTRUCTION]
</instruction>
<answer>
[OUTPUT1]
</answer>
```

FIRST, provide a one-sentence explanation of your evaluation, detailing the reasoning behind your score.
SECOND, on a new line, state only the score on a scale from 0 to 10, where a higher score indicates better overall performance. Your response should follow this format:

```
Explanation: <one-sentence explanation>
Score: <a scale from 0 to 10>
```

---

## H.2  HEAD-TO-HEAD COMPARISON PROMPT

---

**Prompt in Head-to-head Comparison**

Given the question in `[LANG]` language. You are a helpful and precise assistant for checking the quality of the answer.

```
<instruction>
[INSTRUCTION]
</instruction>
<answer1>
[OUTPUT1]
</answer1>
<answer2>
[OUTPUT2]
</answer2>
```

A good answer should follow these rules:

1. It should be in `[LANG]`, unless the instruction explicitly requests a different language.
2. It should be helpful, relevant, detailed and accurate.
3. It should answer the request in the instruction

Please evaluate both answers with your justification, and only provide a score ranging from 0 to 10 after your justifications, the score must be an integer. The score for answer 1 should be wrapped by `<score1>` and `</score1>`, and the score for answer 2 should be wrapped by `<score2>` and `</score2>`.

---

## H.3   X-ALPACAEVAL PROMPT

> **Prompt modified with `weighted_alpaca_eval_gpt4_turbo` in AlpacaEval 2.**
>
> ```
> <|im_start|>system
> ```
> You are a highly efficient assistant, who evaluates and selects the best large language model (LLMs) based on the quality of their responses to a given instruction. This process will be used to create a leaderboard reflecting the most accurate and human-preferred answers.
> ```
> <|im_end|>
> <|im_start|>user
> ```
> I require a leaderboard for various large language models. I'll provide you with prompts given to these models and their corresponding outputs. Your task is to assess these responses, and select the model that produces the best output from a human perspective.
>
> ## Instruction
>
> ```
> {
>     "instruction": """{instruction}""",
> }
> ```
>
> ## Model Outputs
>
> Here are the unordered outputs from the models.  Each output is associated with a specific model, identified by a unique model identifier.
>
> ```
> {
>     {
>         "model_identifier": "m",
>         "output": """{output_1}"""
>     },
>     {
>         "model_identifier": "M",
>         "output": """{output_2}"""
>     }
> }
> ```
>
> ## Task
>
> A good output should be in the same language as the instruction, except when the instruction explicitly requests the output in a different language. Evaluate the models based on the quality and relevance of their outputs, and select the model that generated the best output. Answer by providing the model identifier of the best model. We will use your output as the name of the best model, so make sure your output only contains one of the following model identifiers and nothing else (no quotes, no spaces, no new lines, ...): m or M.
>
> ## Best Model Identifier
> ```
> <|im_end|>
> ```

## H.4   SELF TRANSLATION PROMPT

> **Prompt in Self Translation**
>
> Please translate the following sentences into `[LANGUAGE]`. The input sentences are wrapped by `<sentence>` and `</sentence>`:
>
> ```
> <sentence>
> [TEXT]
> </sentence>
> ```
>
> The translated result should be wrapped by `<translated>` and `</translated>`.

## H.5  MULTILINGUAL REASONING PROMPT

> **Prompt in Multilingual Reasoning**
>
> Below is an instruction that describes a task. Write a response that appropriately completes the request in `[LANGUAGE]`. Please answer in `[LANGUAGE]`.
>
> ### Instruction:
> `[INSTRUCTION]`
>
> ### Response:

