# OpenReview forum: "Language Imbalance Driven Rewarding for Multilingual Self-improving"
_ICLR.cc/2025/Conference — ICLR 2025 Poster_

### Official Review · Reviewer_n8tX · 2024-10-28

**Soundness:** 2
**Presentation:** 3
**Contribution:** 3
**Rating:** 6
**Confidence:** 4

**Summary:**

This paper leverage the langauge imbalance of LLMs to bootstrap the multilingual capabilities of LLM in a self-improving manner. The proposed approach involves training through Iterative Direct Preference Optimization (DPO), which leverages preference data generated by the model itself. By translating prompts and responses between dominant and non-dominant languages, the model assigns preferences, with responses in dominant languages generally preferred. The system iteratively fine-tunes the model on these preferences, improving multilingual performance without human-annotated data. Experiments on the Meta-Llama-3-8B-Instruct model show substantial performance improvements across both instruction-following and arithmetic reasoning tasks, with enhancements across all evaluated languages.

**Strengths:**

1. This work suggests a promising direction for self-improving multilingual LLMs by leveraging intrinsic language imbalance.
2. Experimental results demonstrate significant improvements over baseline models on multilingual benchmarks.
3. The method is clearly explained, which is easy to understand.

**Weaknesses:**

1. The evaluation heavily relies on GPT-4 to assess response quality, but its reliability as a multilingual judge is uncertain, which was discuessed in [1]. The evaluation in Table 1 across various languages may be biased because it's unclear if GPT-4 rates identical responses equally in different languages. A fairer method would be to evaluate responses in the same language, including both directly generated and translated ones, to ensure scores are more comparable.
2. The method depends on LLMs to self-translate responses, but the quality may be inferior to established systems like Google Translate and NLLB, particularly for low-resource languages, as discuessed in [2]. To enhance the analysis, it would be beneficial to assess the method using low-resource languages such as Swahili, Thai, and Bengali, as well as comparing it with more translation systems like Google Translate.

[1] Are Large Language Model-based Evaluators the Solution to Scaling Up Multilingual Evaluation?
[2] Is Translation All You Need? A Study on Solving Multilingual Tasks with Large Language Models.

**Questions:**

1. It would be better to use more advanced LLMs like GPT-4o as the judge due to its stonger multilingual capabilities.
2. Does the approach for weaker models like Llama-2-7B and lower-resource langauges? If it works, this method would be more impactful.

---

> ### Author Response · Authors · 2024-11-20
> **Response to Reviewer n8tX (Part 1)**
>
> Thanks for your efforts to provide insightful comments. We will address your concerns point by point.
>
> > W1 & Q1: The concern about the evaluation's heavy reliance on GPT-4 for assessing response quality
>
> Please see `General Response about GPT-4 on Cross-Language Evaluation Bias`.
>
> > W2: The comparison between self-translate and translation systems using low-resource languages
>
> Thanks for your insightful question! Below, we address why our method prioritizes self-translation and when external translation systems could be considered.
>
> **Perspective 1: Why do we prioritize self-translation over external translation systems?**
>
> + **Self-Improving Paradigm**: Our method is designed as a self-improving paradigm, where the model iteratively refines its capabilities. Introducing external translation systems disrupts this iterative cycle, making it harder to achieve the intended self-improvement and model autonomy.
>
> + **Consistency**: Self-translation aligns closely with the model's inherent generation distribution. This consistency prevents collapse during DPO training, as the model is optimizing against outputs it understands well. In contrast, external translation systems produce distributions that differ from the model's original generation distribution, which may hinder effective training.
>
> + **Efficiency**: Self-translation eliminates the integration complexity associated with third-party systems, making the approach more efficient and streamlined for implementation.
>
> **Perspective 2: When do we use external translation systems?**
>
> The primary goal of using self-translation is to preserve the integrity of the self-improving paradigm. However, in cases where the model's generation capabilities are particularly limited for certain low-resource languages, relaxing this constraint and using an external translator is also a viable solution.
>
> In such scenarios, external translation systems can be leveraged for mutual translation between dominant and low-resource languages to bootstrap performance. However, this would be **a complementary strategy rather than the core of our proposed approach**.
>
>
> **Perspective 3: External Translation Systems as a Complementary Strategy for Low-Resource Languages.**
>
> It is important to note that the effectiveness of post-training on these low-resource languages is inherently limited. The model's multilingual capabilities are primarily developed during the pre-training phase, where it learns from a diverse and extensive multilingual corpus. As such, the gains from post-training are incremental and cannot fully overcome the limitations of the pre-training data for low-resource languages.
>
> Llama3-8b-Instruct demonstrates weak performance in low-resource languages such as Swahili, Thai, and Bengali. To assess the impact of post-training on these languages, we conducted experiments using Llama3-8b-Instruct as the base model, comparing the results with those from the external translation system.
>
> Table n1: The X-AlpacaEval Leaderboard evaluated by `the latest GPT-4o`, with **external Google translate system**.
> | Model                         | en    | bn    | sw    | th    |
> |-------------------------------|-------|-------|-------|-------|
> | Meta-Llama-3-8B-Instruct(M0)   | 35.12% | 1.05%  | 1.01%  | 2.79%  |
> | Iteration1(M1)                 | 38.35% | 4.32%  | 2.98%  | 26.94% |
> | Iteration2(M2)                 | 38.38% | 5.99%  | 3.45%  | 29.10% |
>
> Table n2: The X-AlpacaEval Leaderboard evaluated by `the latest GPT-4o`, with **model performing self-translation**.
> | Model                         | en    | bn    | sw    | th    |
> |-------------------------------|-------|-------|-------|-------|
> | Meta-Llama-3-8B-Instruct(M0)   | 35.12% | 1.05% | 1.01% | 2.79% |
> | Iteration1(M1)                 | 38.22% | 4.06% | 1.15% | 23.27% |
> | Iteration2(M2)                 | 39.27% | 4.49% | 2.07% | 28.07% |
>
>
> `Table n1` and `Table n2` present the results for Google Translate and self-translation, respectively. We summarize the following three points:
>
> 1. Even though the model performs weakly in these languages, our approach remains **effective in low-resource settings** and can iteratively improve the model's performance across all languages.
> 2. Self-translation more effectively improves the performance of English because it avoids introducing external translations, maintaining **a consistent generation space**. This prevents disruption to English's established capabilities, leading to better performance improvements.
> 3. The external Google translation system provides **higher-quality data for low-resource languages**, enhancing the model's capabilities in these languages during optimization due to the model's initially weaker generation capabilities in these languages.
>
> **By prioritizing self-translation while leaving room for the targeted use of external translation systems in extreme cases**, our method aims to balance robustness, efficiency, and the ideal self-improving paradigm.

---

> ### Author Response · Authors · 2024-11-20
> **Response to Reviewer n8tX (Part 2)**
>
> > Q2: Does the approach for weaker models like Llama-2-7B and lower-resource langauges?
>
> In response to the reviewer's inquiry, our experiments demonstrate that the proposed approach does indeed work for both weaker models (like Llama-2-7B) and lower-resource languages (Swahili, Thai, and Bengali). Specifically:
>
> > The Supplementary experiments on the weaker model, Llama-2-7b-Chat as the base model.
>
> Table n3: The X-AlpacaEval Leaderboard evaluated by `the latest GPT-4o`.
> | Model            | en     | es     | ru     | de     | fr     | Avg   |
> |------------------|--------|--------|--------|--------|--------|-------|
> | Llama2-7B-Chat (M0)   | 11.60% | 3.40%  | 0.32%  | 0.87%  | 0.69%  | 3.38% |
> | Iteration1 (M1)  | 14.62% | 5.30%  | 1.91%  | 1.89%  | 2.68%  | 5.28% |
> | Iteration2 (M2)  | 14.86% | 6.62%  | 4.51%  | 3.62%  | 6.35%  | 7.19% |
>
>
> `Table n3` demonstrates that even when starting with **a model with weaker multilingual capabilities, such as Llama2-7B-Chat**, which exhibits extremely low performance in languages like Russian (ru), German (de), and French (fr) on the X-AlpacaEval, **significant improvements can be achieved**.
>
> By leveraging **language imbalance driven rewarding** for self-multilingual optimization across two iterations, the model shows substantial enhancement across all training languages, particularly in those where the original model's performance was initially weaker.
>
> > The Supplementary experiments on lower resource languages (bn, th, sw), Llama-3-8b-Instruct as base model.
>
> Detailed results and analysis for these languages can be found in `W2`.
>
> We are more than willing to continue engaging in in-depth discussions. If you have any further questions, please feel free to comment at any time.

---

> > ### Comment · Reviewer_n8tX · 2024-11-26
> >
> > Thanks to the authors for the clarification and the additional experiment. Currently, my main concern is that this work's core contribution is similar to that of MAPO, which diminishes its novelty. Therefore, I decide to maintain my scores.

---

> > > ### Author Response · Authors · 2024-11-26
> > > **Response to Reviewer’s Concern: Key Differences from MAPO and the Novelty of Our Approach**
> > >
> > > **Thank you for your quick response!** We appreciate your feedback regarding the core contribution of our work, particularly regarding its similarity to MAPO. We would like to clarify key aspects of our approach, **which we believe distinguish it from MAPO, and highlight its unique contribution to the field.**
> > >
> > > >**Motivation and Key Differences Between Our Work and MAPO**
> > >
> > > Our work focuses on **multilingual self-rewarding**—a self-improvement mechanism that enhances the general multilingual capabilities of large language models (LLMs) without relying on external resources. In contrast, MAPO primarily addresses **multilingual mathematical reasoning** tasks and employs external machine translation (MT) models to create reward signals.
> > >
> > > In our manuscript, we explicitly highlight the inspiration drawn from **Self-rewarding** (Yuan et al., ICML 2024), as detailed in **Lines 53-55**. To clarify the relationship and distinctions between our method, Self-rewarding, and MAPO, we provide the following table (Table r1):
> > >
> > > | **Aspect**                      | **Ours**                                       | **Self-rewarding**                          | **MAPO**                                      |
> > > |----------------------------------|------------------------------------------------|---------------------------------------------|-----------------------------------------------|
> > > | **Task Focus**                   | **Instruction following**, _math reasoning_    | **Instruction following**                   | _Math reasoning_                              |
> > > | **Training Method**              | **Iterative DPO**                              | **Iterative DPO**                           | **Iterative DPO** or PPO                     |
> > > | **Reward Signal**                | Intrinsic language imbalance                   | LLM-as-a-Judge (self-judged)                | Semantic consistency via external MT model    |
> > > | **Preference Data**              | **Preference pairs**                           | **Preference pairs**                        | Preference score from MT model                |
> > > | **External Model/Data**          | **None**                                       | **None**                                    | Off-the-shelf MT model                        |
> > > | **Multilingual**                 | _Yes_                                          | No                                          | _Yes_                                         |
> > >
> > > As illustrated in **Table r1**, our method shares more similarities with Self-rewarding (highlighted in **bold**) than with MAPO. We use the same tasks (instruction following), and the same training method (Iterative DPO), and construct preference pairs in a similar way (via self-rewarding). **The only overlap with MAPO is the use of multilingual mathematical reasoning tasks, but this is where the similarities end.**
> > >
> > > >**Key Differences from MAPO**
> > >
> > > 1. **MT Utilization**:
> > >    - **Our approach** uses **self-translation** of responses across languages, which allows for multilingual preference optimization without relying on any external models.
> > >    - **MAPO**, on the other hand, employs an **external MT model** to evaluate and score consistency between reasoning paths in different languages. This distinction is crucial, as we do not use external models to generate reward signals.
> > >
> > > 2. **Preference Pair Construction**:
> > >    - **In our method**, preference pairs are constructed based on the **language imbalance** prior (dominant languages tend to outperform non-dominant languages).
> > >    - **MAPO** constructs preference pairs **indirectly**, relying on the **reward scores derived from an external MT model** to evaluate the semantic consistency between reasoning paths across languages.
> > >    - Additionally, our method can construct preference pairs for **both dominant and non-dominant languages**, whereas MAPO is **limited to non-dominant languages** due to the use of external MT systems.
> > >
> > > 3. **Training Paradigm**:
> > >    - **Our approach** exclusively uses **Iterative DPO without any external supervision**, aligning with the self-improving paradigm.
> > >    - **MAPO** uses both **DPO and PPO**. The use of PPO is a trade-off for better training efficiency, as it operates on the **reward scores generated by the external MT model**.
> > >
> > > >**Clarification on the Core Contribution and Novelty of Our Work**
> > >
> > > We hope that the clarifications above address your concerns about the novelty of our approach. We believe that the **self-rewarding mechanism based on language imbalance**, coupled with **LLM self-translation**, presents **a novel and meaningful contribution** to **the field of multilingual self-improvement**. We look forward to receiving your positive evaluation!
> > >
> > > If you have any further questions, please feel free to comment at any time. We hope the analysis provided above helps you gain a deeper understanding of the core contribution and novelty of our work. Looking forward to your reply!

---

> > > ### Author Response · Authors · 2024-11-29
> > > **Update Further response on Novelty and Language Bias in LLM-as-a-Judge, Looking Forward to Your Reply!**
> > >
> > > Dear Reviewer `n8tX`,
> > >
> > > Thank you for your time and effort in reviewing our work. We would like to further address two key points:
> > >
> > > + We have provided a detailed explanation of **the key differences between our approach and MAPO**, and the novelty of our method has been recognized by Reviewer `w67r`. As Reviewer `w67r` noted, **"looking at the intrinsic capability difference is still a new method"**. We believe that the **self-rewarding mechanism based on language imbalance**, coupled with LLM self-translation, offers **a novel and meaningful contribution** to the field of **multilingual self-improvement**. Moreover, we have revised the manuscript to include a more detailed discussion of related works (`highlighted in blue`), further clarifying the contributions of our approach. We would greatly appreciate it if you could kindly reconsider the novelty of our work in light of the detailed explanations provided above!
> > >
> > > + Additionally, we have expanded on the issue of language bias in LLM-as-a-Judge in **General Response about GPT-4 on Cross-Language Evaluation Bias**. We hope this provides a deeper understanding of how to mitigate language-specific biases in the LLM-as-a-Judge within our work.
> > >
> > > **We look forward to hearing your thoughts and feedback on these explanations, and we hope to receive your positive evaluation of Language Imbalance-driven Rewarding. Thank you again for your valuable feedback. We look forward to your reply. Wish you have a nice day!**
> > >
> > > Best wishes

---

> > > > ### Comment · Reviewer_n8tX · 2024-12-03
> > > >
> > > > Thank you for outlining the similarities and differences between this work and MAPO. After revisiting the MAPO paper and considering the feedback from other reviewers, I would like to clarify the following points:
> > > >
> > > > > Main similarities with MAPO
> > > >
> > > > Both this work and MAPO leverage the language imbalance to optimize the models.
> > > >
> > > > > Main differences with MAPO
> > > >
> > > > MAPO constructs preference pairs by calculating the alignment scores with dominant languages. The responses with higher alignment scores will be preferred. This work generates preference pairs solely by translation. The responses in dominant languages or translated from dominant languages are preferences.
> > > >
> > > > > Other minor differences
> > > >
> > > > 1. This work improves the performance of both English and other languages. (Advantage)
> > > > 2. This work uses self-translation instead of an external translation model. (does not matter)
> > > > 3. This work applies to both reasoning tasks and more general tasks. (Advantage)
> > > > 4. Training methods are different. (does not matter)
> > > >
> > > > > What is missing?
> > > >
> > > > Since both this work and MAPO leverage the language imbalance and can be applied to reasoning tasks, it is important to compare their effectiveness on tasks like MGSM, as mentioned by Reviewer AMKf. This will make the work more comprehensive and convincing.
> > > >
> > > > > Conclusion
> > > >
> > > > Even though the work has similar motivations as MAPO, it has some new contributions. Therefore, I would like to increase the score.

---

> > > > > ### Author Response · Authors · 2024-12-03
> > > > > **Thank you for raising the score and recognizing our contributions！**
> > > > >
> > > > > We sincerely appreciate your thoughtful feedback and the increase in the score. Your recognition of our contributions means a great deal to our work! We will conduct experiments on MAPO for the arithmetic reasoning task. However, as the discussion phase is nearing its end, we regret that we may not be able to complete the experiments in time. Nevertheless, we will provide the results as soon as possible. Once again, thank you very much for your efforts to help improve our work. Wish you have a nice day! :）

---

> ### Author Response · Authors · 2024-11-25
> **Looking Forward to Further Discussion**
>
> Dear reviewer n8tX,
>
> As the discussion phase is nearing its conclusion, we would like to know if our responses have addressed your concerns. Looking forward to further discussion with you. We are happy to address any questions if such occur. Wish you have a nice day!
>
> Best regards,
>
> Authors

---

### Official Review · Reviewer_w67r · 2024-11-02

**Soundness:** 3
**Presentation:** 3
**Contribution:** 3
**Rating:** 6
**Confidence:** 4

**Summary:**

This work proposes to create a preference dataset using a multilingual LLM by exploiting the quality difference in responses in dominant languages (`dl`) and non-dominant languages (`nl`). The paper makes two assumptions that for the same instruction, 1) an `nl` response after being translated to `dl` is still worse than the original `dl` response, and similarly, 2) a `dl` response after being translated to `nl` is still better than the original `nl` response. The authors then provided empirical evidence for these assumptions. Iterative DPO training was done using the synthesized data and the trained models were tested on multilingual AlpacaEval, multilingual MT-bench, and some structured multilingual tasks. Results are shown to be in favour of the proposed method that models can iteratively improve by using data synthesized from itself.

**Strengths:**

- I think the idea of exploiting the gap in an LLM's inherent language capability for self-improvement is interesting and intuitive.

- The paper has extensive experiments across open-ended and close-ended benchmarks. Results are consistent and in favour of the proposed method. It is clear that through iterative data synthesis and training, models can progressively improve in both dominant languages (`dl`) and non-dominant languages (`nl`).

**Weaknesses:**

1. Several factors in asserting the assumptions could not be carefully controlled. Table 1 line 180: GPT4-as-a-judge is used to confirm the quality of responses in different languages, however, there is no guarantee that the scores for different languages are on the same scale and that GPT4 is able to judge those languages using the same "standard". This could apply to Table 2 and Table 3 too.

2. I find the motivation of creating (`nl`, `dl->nl`) preference data reasonable. However, I did not find creating the response pair (`dl`, `nl->dl`) intuitive. Since the model already could generate `dl` well (higher quality), why would having a lower-quality sample benefit alignment, especially as your Finding 2 that "the dominant language also benefits [...] a phenomenon that has rarely been observed"? Do you have any explanations for this?

3. I think that all multilingual benchmarks used in this work, both open-ended and structure tests, are machine-translated from English to other languages. Since the preference data is also constructed using self-translation e.g. (`nl`, `dl->nl`), it might be possible that some kind of "translationese bias" is exploited as opposed to actually improve multilingual capability.

**Questions:**

1. Table 2 row 2 "Self Translation" is confusing me: which language is translated into English (8.03)?
2. The structure of section 5 looks weird. Why would arithmetic reasoning be a separate section/experiment?
3. Line 143, Equation (7), the length normalization term $|y_{win}|$ is inside $log$. Is this correct?

---

> ### Author Response · Authors · 2024-11-20
> **Response to Reviewer w67r**
>
> Thanks for your insightful questions and we believe they hold significant value for our work. We try to resolve your concerns below.
>
> > W1: The quality of GPT-as-a-judge in different languages
>
> Please refer to the `General Response about GPT-4 on Cross-Language Evaluation Bias` for further details.
>
> > W2: The explanation of the benefits of incorporating negative samples to construct English preference pairs for alignment.
>
> In the preference data for English, we construct preference pairs by randomly sampling responses from other languages and translating them into English to serve as negative samples. We argue that using negative samples to build preference pairs for DPO training helps the model penalize the generation of negative samples, a process similar to machine unlearning [r1]. This approach also leads to improved model performance.
>
> Experimental results in `Figure 2 and Table 4 of the manuscript`, focusing on the dominant language, show that incorporating negative samples in preference pair construction significantly improves the model's performance. This finding aligns with concurrent work presented at EMNLP 2024 [r2], which also demonstrates that alignment can be achieved using solely negative samples.
>
> > W3: The concerns about "translationese bias" in multilingual benchmarks evaluation
>
> The use of machine-translated benchmarks can sometimes introduce artifacts commonly associated with "translationese", such as simplified syntax, lexical choices closer to the source language, or other translation-induced patterns. However, our approach leverages LLMs for self-translation to construct training data, which offers key advantages:
>
> + **Different Data Distributions**: Our method uses LLM self-translation to construct training data, while multilingual benchmarks are derived from machine translation of English datasets. This ensures that the training data and benchmark data have different distributions, effectively minimizing the risk of translationese bias influencing performance.
>
> + **Reduction of Translationese Artifacts**: LLM self-translation significantly reduces translationese effects, producing fluent and natural translations that align closely with native text. This is supported by prior works ([r3, r4]), which highlight the high-quality outputs of LLMs.
>
> + **Robustness Across Tasks**: The method's consistent improvements across open-ended (e.g., X-AlpacaEval, MTBench, and Reasoning) tasks and NLU (e.g., MMLU, HellaSwag, ARC challenge, and TruthfulQA) tasks demonstrate genuine multilingual capability rather than reliance on translation-specific patterns.
>
> > Q1: Which language is translated into English
>
> To ensure the diversity of negative samples in English preference pairs, we randomly sample responses from non-dominant languages and translate them into English.
>
> > Q2: Why would arithmetic reasoning be a separate section/experiment?
>
> 1. By separating this experiment, we aim to evaluate the model's performance on the reasoning task, providing a clearer understanding of its strengths and limitations beyond the general instruction-following task. This separation allows for a more targeted analysis of how the "Language Imbalance Driven Rewarding" method performs on arithmetic reasoning task, which may differ from general instruction following task.
>
> 2. The dataset, evaluation, and metrics used in the Arithmetic Reasoning task differ significantly from those in the General Instruction-following task. To make the experimental section clearer, we have divided the experiments into two separate sections.
>
> > Q3: Typo Error in Eq(7).
>
> Thank you for pointing out the error in the formula notation. We have corrected these in the revised version.
>
>
> We hope the above clarifications have enhanced your understanding of our work. We appreciate your valuable feedback and are eager to discuss any additional questions or comments further.
>
> ---
> [r1] Zhang, Ruiqi, et al. "Negative preference optimization: From catastrophic collapse to effective unlearning." arXiv preprint arXiv:2404.05868 (2024).
>
> [r2] Duan, Shitong, et al. "Negating Negatives: Alignment with Human Negative Samples via Distributional Dispreference Optimization." Findings of the Association for Computational Linguistics: EMNLP 2024. 2024.
>
> [r3] Chen, Pinzhen, et al. "Iterative translation refinement with large language models." arXiv preprint arXiv:2306.03856 (2023).
>
> [r4] Kunilovskaya, Maria, et al. "Mitigating Translationese with GPT-4: Strategies and Performance." Proceedings of the 25th Annual Conference of the European Association for Machine Translation (Volume 1). 2024.

---

> ### Author Response · Authors · 2024-11-25
> **Looking Forward to Further Discussion**
>
> Dear reviewer w67r,
>
> As the discussion phase is nearing its conclusion, we would like to know if our responses have addressed your concerns. Looking forward to further discussion with you. We are happy to address any questions if such occur. Wish you have a nice day!
>
> Best regards,
>
> Authors

---

> ### Comment · Reviewer_w67r · 2024-11-26
> **Reviewer's comment after rebuttal. Adding some comments + maintaining score 5.**
>
> Thank you to the authors for additional information. I have carefully checked your response to my review as well as other reviewers' comments. Below are my thoughts.
>
> - **W1**: it is not addressed because my concern is that LLM-as-a-judge cannot score different languages with the same "standard" or even score range. E.g. for the same high-quality response/content, you may get an 8 for English and a 7 for Chinese, just due to the judge's score bias. This is also raised by **Reviewer n8tX**. I think your current practice of adding more judges cannot prove or disprove this. Perhaps one thing you could consider is that:
>     - 1) for the same question in different languages, create the same response/content but in different languages. Note that these questions should test for objective and factual knowledge and cannot have language/culture-dependent answers (you may need more constraints too to carefully control this)
>     - 2) show that judges can award the same scores to those responses in different languages.
> ----
> - **W2**, **W3**, **Q1**, and **Q3** are addressed.
> ----
> - **Q2**: I still think the writing structure is weird but I respect your choice.
> ----
> - **Novelty**: The other two reviewers raised issues with originality. This is not the first paper that exploits/explores language capability differences as a reward signal, but I think looking at the intrinsic capability difference is still a new method. Whilst certainly not plagiarism, authors should discuss other works more thoroughly.

---

> > ### Author Response · Authors · 2024-11-27
> > **Response to Reviewer’s Concerns: Language Bias in LLM-as-a-Judge**
> >
> > Thank you for your thoughtful feedback and for carefully considering our response to your review, as well as the comments from other reviewers. We appreciate your insights and would like to address the points you raised.
> >
> > > **W1: Language Bias in LLM-as-a-Judge**
> >
> > In `Table 1 of the manuscript`, we evaluate the GPT score for individual responses in different languages, e.g., $y^{en}$ or $y^{ru}$. This approach may introduce language bias, as scores could vary depending on the language in which the response is generated. To mitigate this potential issue, we perform more detailed pairwise comparisons within the same language in `Tables 2 and 3 of the manuscript`, thereby avoiding cross-lingual scoring bias. Importantly, this language bias in `Table 1` does not affect our conclusion, as we focus on constructing pairwise comparisons between responses within the same language for `Tables 2 and 3`.
> >
> >
> > Specifically, in each column of `Table 2`, both responses are in the same language. Take **'ru'** as an example. The **'Self Generation'** GPT score is calculated on $y^{ru}$ and the **'Self Translation'** GPT score is calculated on $y^{en \to ru}$. As both responses are in **'ru'**, it does not introduce potential language bias of LLM-as-a-Judge, providing a fair comparison. Similarly, in `Table 3`, the reward accuracy is evaluated on preference pairs consisting of $y^{ru}$ and $y^{en \to ru}$, which also avoids the cross-lingual comparison issue.
> >
> > Both `Tables 2 and 3` evaluate GPT score based on **pairwise comparisons within the same language**. This methodology inherently **avoids the cross-lingual comparison issue**, ensuring **a fairer and more consistent assessment** since all evaluations are conducted within the same language.
> >
> > **Summary:**
> >
> > - In `Table 1`, GPT scores are assigned to individual responses in different languages, this could introduce language bias. However, this does not affect our conclusion.
> > - In `Tables 2 and 3`, pairwise comparisons are conducted **within the same language**, making the evaluation **fairer and more consistent**.
> > - By translating responses into the same language, we create preference pairs that **avoid language-specific biases** and focus on the **relative quality** of the responses.
> >
> > Our approach, using the LLM-as-a-judge, ensures that the evaluation of responses is **consistent** and **unbiased**, as all comparisons are done within the **same language**.
> >
> > > **W2, W3, Q1, and Q3**
> >
> > We are glad to hear that our responses address your concerns and appreciate your constructive feedback.
> >
> > > **Q2: Writing Structure**
> >
> > We appreciate your constructive feedback about the structure and will review the manuscript to improve clarity and readability.
> >
> > > **Novelty**
> >
> > **Thank you very much for recognizing the novelty of our approach. We appreciate your recognition that our focus on intrinsic language capability differences offers a novel perspective**. We believe that the self-rewarding mechanism based on language imbalance, coupled with LLM self-translation, presents a novel and meaningful contribution to the field of multilingual self-improvement. We have revised the manuscript to include a more detailed discussion of related works `(highlighted in blue)`, further clarifying the contributions of our approach.
> >
> > Thank you again for your thoughtful review. We look forward to your reply! Have a nice day!

---

> > > ### Comment · Reviewer_w67r · 2024-11-27
> > > **Thanks, got it. Will raise score to 6.**
> > >
> > > Right, I now understand that `Table 2` provides some level of control of cross-lingual scoring (range) bias. This resolves my problem with the underlying assumption. I have raised the score to 6.

---

> > > > ### Author Response · Authors · 2024-11-27
> > > > **Thank you for raising the score!**
> > > >
> > > > Dear Reviewer w67r,
> > > >
> > > > Thank you for raising the score! We sincerely appreciate the valuable feedback you provided, which has been incredibly helpful in improving our work.
> > > >
> > > > Best,
> > > >
> > > > Authors

---

### Official Review · Reviewer_AMKf · 2024-11-05

**Soundness:** 2
**Presentation:** 3
**Contribution:** 2
**Rating:** 3
**Confidence:** 4

**Summary:**

This paper presents a method for bridging the language abilities for dominant languages and non-dominant languages. The method use the translation between the generation results in the dominant language and non-dominant language to build preference pairs, and conduct an iterative DPO optimization. The experiment shows improvement over the original model.

**Strengths:**

The method is effective in improving the performance of general instruction following and mathematical reasoning.

The paper also presents analysis for the effectiveness and accuracy of the preference pairs, which serves as a nice support for the method.

**Weaknesses:**

The paper is not the first attempt to improve the multilingual ability of LLMs by cross-lingual optimizations. In one of the cited paper, She et al., ACL 2024, where experiments are conducted in optimizing the preference with DPO and an off-the-shell translator.

I find the proposed method quite similar to the above one, but in this paper there is no clear indication of the potential relations.

**Questions:**

I am not sure why languages such as es and fr only have a reward accuracy of 60%. It seems that they are well represented in the training data. It would be helpful if further explanation or analysis are provided.

**Details Of Ethics Concerns:**

As I mentioned in the weaknesses, the paper present a method that use translation to build multilingual preferences, and use DPO to optimize the model, which is quite similar to the one proposed in She et al., ACL 2024. The authors obviously knew the referenced paper, but provides no explanation in the relation between the two. I don't think this is inappropriate. In case I was personally biased, I raise the problem to the ethics committee for further judgement.

---

> ### Author Response · Authors · 2024-11-13
> **Clarification on Ethics Concerns**
>
> We would like to emphasize that the focus of our method is multilingual self-rewarding that self-improves the general multilingual capability without external resources. In `Lines 53-55`, we explicitly mention that our work is primarily inspired by Self-rewarding (Yuan et al., ICML 2024). Table r1 provides a detailed comparison, highlighting the potential similarities and differences between our work, Self-rewarding, and MAPO (She et al., ACL 2024).
>
>
> Table r1: The relationship between our approach, Self-rewarding, and MAPO.
> |                     | Ours                                                             | Self-rewarding              | MAPO                                                 |
> |---------------------|------------------------------------------------------------------|-----------------------------|------------------------------------------------------|
> | Task                | **Instruction following (mainly)** and _math reasoning (additionally)_ | **Instruction following**       | _Math reasoning_                                       |
> | Training method     | **Iterative DPO**                                              | **Iterative DPO**               | **Iterative DPO** or PPO                               |
> | Reward signal (core contribution)       | Intrinsic language imbalance (prior)                                    | LLM-as-a-Judge (self-judge) | Semantic consistency (evaluated by external MT model)         |
> | Preference data form     | **Preference pairs**                                                 | **Preference pairs**            | Preference score ('alignment' score) for each sample |
> | External model/data | **None**                                                             | **None**                        | Off-the-shelf MT model                               |
> | Multilingual    | _Yes_                                                              | No                          | _Yes_                                                  |
>
> As shown in `Table r1`, our approach bears more resemblance to Self-rewarding (highlighted in **bold**), including the task we focused on, the training method, preference data form, and the self-improving paradigm. Therefore, we mainly contextualized our approach with LLM self-improving methods in related works. In contrast, the only similarity between our work and MAPO is that we all experimented on multilingual mathematical reasoning task (the _italicized_ text).
>
> Regarding the reviewer’s comment on the similarity between our approach and MAPO, particularly concerning the use of MT for building preference data and iterative DPO for optimization, we believe there are key differences that set our work apart.
>
> 1. **MT utilization**: Our work only leverages the **LLM itself** to translate responses across languages, while MAPO employs an **external MT model** as a reward model.
> 2. **Preference pair construction**:
>
>     a) In our work, preference pairs are constructed based on **language imbalance**—specifically, the prior that dominant languages generally outperform non-dominant ones. This contrasts with MAPO, which does not directly construct preference pairs. Instead, MAPO first employs an MT model to assess the **consistency between reasoning paths** generated in different languages, where the MT model functions as a reward model. The reward score derived from this consistency assessment is then used to construct preference pairs.
>
>     b) Our approach could construct preference pairs for **both dominate and non-dominate languages**, while MAPO can only derive reward signal for **non-dominate ones**.
> 4. **Iterative DPO training**: As shown in `Table r1`, we draw inspiration from the Self-rewarding method, which constructs preference pairs without external supervision and **only adopts iterative DPO training**. MAPO, on the other hand, **employs both PPO and DPO training** in their experiment, as they can obtain a reward score ('alignment' score) for each response using the external MT model.
>
> In conclusion, we believe the similarity in `Table r1` and the distinctions listed above make it clear that **our approach is directly inspired by Self-rewarding instead of MAPO**.
>
> We did not include MAPO as a major related work, since it is designed for only reasoning tasks rather than general ones such as instruction following tasks. This is because off-the-shelf translators fail to assess consistency between LLM responses due to their limited context size. Besides, it does not completely obey the self-improving paradigm, due to the use of the external translator.
>
> To clear up the misunderstandings, we will include a detailed discussion in the revised version on the relationship between our approach, Self-rewarding, and MAPO, emphasizing their major differences.
>
> We hope the evidence provided above could address your concerns. We would greatly appreciate it if you could kindly reconsider our work with these distinctions in mind.

---

> ### Author Response · Authors · 2024-11-14
> **Response to Reviewer AMKf**
>
> We sincerely appreciate your questions. We highly value your feedback and provide detailed explanations to address your concerns.
>
> > Q1: Explanation and Analysis of the relation between Language performance and reward accuracy.
>
> In `Lines 164-165`, we claimed that the insight behind our work is Language Imbalance Driven Rewarding: **The greater the disparity between the capabilities of the dominant and non-dominant languages, the stronger the reward signal, which subsequently leads to higher reward accuracy**.
>
> Therefore, (es, fr) are **well represented** in the training data, resulting in **a narrow performance gap with English** in preference pairs. As a result, the reward signal becomes **weaker**, leading to **lower reward accuracy**.
>
> Here is a more detailed analysis, corresponding to `Section 3` of the manuscript：
>
> In `Table 1`, we can observe that the GPT-4 score for responses in (es, fr, it) are noticeably higher compared to those in (de, ja, ru), indicating that the former languages are well represented in the training data.
>
> In `Table 2`, we can clearly observe that within the group (es, fr, it), the gap between the responses for "Self Generation" (the model generates responses in non-dominant languages) and "Self Translation" (the model translates English responses into non-dominant languages) is significantly smaller compared to the group (de, ja, ru). This indicates that the GPT-4 score gap between (es, fr, it) and English is narrower, suggesting a weaker reward signal, which results in lower reward accuracy.
>
> The above analysis is consistent with the results in `Table 3`, where we directly obtain the reward accuracy of multilingual preference pairs. The model's weaker performance on (de, ja, ru) creates a large gap with English performance, which results in higher accuracy in preference pairs where English responses are chosen. In contrast, its stronger performance on (es, fr, it) creates a smaller gap with English performance, leading to lower reward accuracy in preference pairs where English responses are chosen.
>
> We hope the above explanations have enhanced your understanding of our work. We appreciate your valuable feedback and are eager to further discuss any additional questions.

---

> ### Comment · Reviewer_AMKf · 2024-11-15
>
> I do recognize the paper has its own merit, for example, in designing the reward signal and using the model itself as the source of preference. However, I don't think it is proper to regard it as a complete separate work from She et al, 2024.
>
> For the record, the authors do not mentioned the technical similarity to She et al., 2024 throughout the paper. The only mention is "existing LLMs demonstrate inconsistent reasoning capabilities across different languages (She et al., 2024)." I don't think this is an acceptable practice.
>
> As additional comments to the authors in the above reply:
>
> 1.  "Ours" uses iterative DPO, "MAPO" uses Iterative DPO or PPO.
>
> 2. "Ours" uses preference pairs, "MAPO" uses preference pairs for the DPO training, but use preference score for PPO training.
>
> How could the above two points been treated as a technical difference between the two work? The attempts in this paper is just a subset of the cited paper.
>
> In my opinion, the differences in techniques is not directly related to the domain or tasks. There are many research in the general task/domain that are inspired by previous attempts in a certain task/domain, or vice versa.
>
> Even for the technical part, the proposed work and She et al., 2024 both use translation as a bridge between the results of the dominant language and non-dominant language, though in slightly different way. I do suggest a comparison between the two, which might bring new insight in utilizing the translation knowledge.

---

> ### Author Response · Authors · 2024-11-20
> **Further Response to Reviewer AMKf**
>
> Thank you for your response. In our initial response, we provide `Table r1` to clarify that **our approach is directly inspired by Self-rewarding instead of MAPO**, to avoid potential concern regarding plagiarism.
>
> After reading your response, we believe there are still misunderstandings to be resolved:
>
> > The differences in constructing preference pairs and the utilization of DPO
>
> The purpose of `Table r1` is to compare our approach with Self-Rewarding and MAPO, highlighting the source of our inspiration. While MAPO also conducts experiments with iterative DPO training using constructed preference pairs, their pairs are **derived from reward scores generated by a reward model**. In contrast, our approach constructs preference pairs **directly based on the prior of language imbalance**, which is a key difference in our methodology.
>
> Ideally, a reward model could always be used to score responses and create preference pairs. When viewed solely from this perspective, all DPO-related works could technically be considered a subset of PPO works. However, we contend that the key consideration should not be this theoretical hierarchy but rather the rationale behind selecting a specific preference optimization method for a given approach.
>
> From this standpoint, our method is clearly **off-policy**, as we train directly on self-translated responses that are not sampled from the model's output distribution for the given instruction. Consequently, our approach exclusively employs DPO training. In contrast, MAPO is **on-policy**, as it scores sampled responses directly and trains on them. MAPO's use of DPO is a trade-off aimed at improving training efficiency rather than a fundamental aspect of its methodology.
>
> > Translation as a bridge in cross-lingual alignment.
>
> Regarding the usage of the MT technique. We mainly utilized MT to translate responses across languages. Such usage is similar to previous cross-lingual alignment works [r1, r2], where the translated responses are utilized to further fine-tune the model. In contrast, MAPO only uses the MT model as a reward model, they do not actually generate translations and use them for model training. Therefore, we believe there is a huge difference in the utilization of MT between our work and MAPO.
>
> > The discussion about MAPO has been added to the revised paper.
>
> We thank the reviewer for pointing out this issue. In the revised version of our manuscript, we have added detailed comparisons and discussions (highlighted in blue) to provide a more thorough analysis of the method and contributions of MAPO.
>
> We hope the above explanations will help you better understand our work. We are happy to communicate with you further for any questions to enhance our manuscript.
>
> ---
> [r1] Zhang, Yuanchi, et al. "Enhancing Multilingual Capabilities of Large Language Models through Self-Distillation from Resource-Rich Languages." arXiv preprint arXiv:2402.12204 (2024).
>
> [r2] Chen, Nuo, et al. "Breaking language barriers in multilingual mathematical reasoning: Insights and observations." arXiv preprint arXiv:2310.20246 (2023).

---

> ### Author Response · Authors · 2024-11-25
> **Looking Forward to Further Discussion**
>
> Dear reviewer AMKf,
>
> As the discussion phase is nearing its conclusion, we would like to know if our responses have addressed your concerns. Looking forward to further discussion with you. We are happy to address any questions if such occur. Wish you have a nice day!
>
> Best wishes!

---

> > ### Comment · Reviewer_AMKf · 2024-11-30
> >
> > I think I have made my point fairly clear.
> >
> > 1. For the issues related to Ethics Concerns:
> >
> > When presenting your work, I think it is necessary to introduce related work and provide necessary background information, which is then useful to build your own contribution to the problem.
> >
> > I think the practice of this paper is improper, because when introduce its own work, it ignores the work of  She et al. ACL 2024, which is quite similar in the framework, the training method, etc.. They are even quite similar in the idea of using "Language Imbalance Driven Rewarding" as the title of this paper.
> >
> > I do notice the authors argued about the differences of the two work in the response. However, the fact that two work are not completely the same does not mean they do not share quite a lot of points.
> >
> > The authors claimed the whole technical part is their own contribution, but inspired by Yuan et al. (without any explicit mention of the similar work She et al.), which I think is a obvious over-claim, and may also be seem as plagiarism.
> >
> > 2. For the response and discussions:
> >
> > I dont think the authors provide necessary comparison and discussion during the response period. As I pointed out in my previous reply, it is ridiculous the authors try to make the point that using iterative DPO instead of (iterative DPO or PPO) is one of the cases that make this work different from She et al. 2023. In this case, the attempts in this paper is just a subset of the cited paper.
> >
> > 3. Technical Point.
> >
> > I agree this paper present a different path of using multilingual signal in assisting the learning of non-dominant language. Although learning from the model itself is interesting, it is important to demonstrate how different paths compared to each other.
> >
> > The authors may argue that they are conducting experiments in the general instruction following settings, however, in section 5 of the revised paper, which is exactly the mGSM dataset previous work has been working on, the author does not provide necessary comparison with She et al. ACL2023.
> >
> > For the above points, I insist on my original decision of this paper. And I encourage my fellow reviewers to read the cited paper to make a final decision.

---

### Author Response · Authors · 2024-11-20
**General Response about GPT-4 on Cross-Language Evaluation Bias**

Thank reviewer `w67r` and `n8tX` for raising the issue about GPT-4 on cross-language evaluation bias. We greatly appreciate your attention to this matter, and below, we address this issue from three perspectives to ensure the reliability of our findings:

> Addressing Cross-Language Evaluation Bias (`Table 1 of the manuscript` and `Table G1`)

We acknowledge the limitation that GPT-4's scores across languages may not be entirely consistent, as its evaluation capabilities can vary depending on the language. To investigate this potential bias, we employed the more advanced GPT-4o model (**gpt-4o-2024-08-06**) for evaluation. Specifically, we used the same responses as those in `Table 1 of the manuscript` and evaluated their quality using GPT-4o. The results in `Table G1` highlight an inherent imbalance in the model's multilingual capabilities, which aligns with the findings from GPT-4 (**gpt-4-1106-preview**) evaluations in `Table 1 of the manuscript`. The consistent results from GPT-4 (`Table 1`) and GPT-4o (`Table G1`) across different evaluation models indicate the robustness of our findings, even in the presence of potential cross-linguistic evaluation biases.

Table G1:  The average quality of responses evaluated by `the latest GPT-4o`.
| Model               | en   | de   | es   | fr   | it   | ja   | ru   |
|---------------------|------|------|------|------|------|------|------|
| Llama-3-8B-Instruct | **9.68** | 8.99 | 7.64 | 7.65 | 6.40 | 2.97 | 4.51 |

> Ensuring Fairness in Same-Language Evaluations (`Table 2 and Table 3 of the manuscript`)

In `Table 2 of the manuscript`, both "Self Generation" and "Self Translation" involve responses in the same language, while in `Table 3 of the manuscript`, preference pairs are similarly constructed within the same language. These evaluations inherently avoid the cross-lingual evaluation issue, providing a relatively fair comparison and ensuring that the evaluation remains consistent and unbiased within the same language.

> Validation on the X-Alpacaeval (`Table 4 of the manuscript` and `Table G2`)

To ensure a fair comparison, we evaluated win rates on the X-AlpacaEval benchmark using GPT-4o (gpt-4o-2024-08-06). However, the existing AlpacaEval repository does not provide a GPT-4o evaluator configuration with human-alignment calibration. As a result, we had to use GPT-4o with a configuration calibrated for GPT-4 Turbo, which may introduce some bias.

The results remain consistent with `Table 4 of the manuscript`, evaluated by GPT-4 Turbo. These findings validate the robustness of our approach and demonstrate consistent multilingual performance improvements, despite potential minor evaluation bias.

Table G2: The X-AlpacaEval Leaderboard evaluated by `the latest GPT-4o`.
| Model                         | en     | es     | ru     | de     | fr     | Avg   |
|-------------------------------|--------|--------|--------|--------|--------|-------|
| Meta-Llama-3-8B-Instruct(M0) | 35.12% | 26.60% | 8.35%  | 9.93%  | 19.21% | 19.84% |
| Iteration1(M1)               | 38.44% | 34.28% | 25.79% | 26.34% | 30.13% | 31.00% |
| Iteration2(M2)               | 39.78% | 33.68% | 28.37% | 26.67% | 31.49% | 32.00% |


By adopting these measures, we believe we have adequately addressed the uncertainty surrounding GPT-4's reliability as a multilingual evaluator, ensuring that the assessments are as fair and consistent as possible. We appreciate the reviewer's suggestion and have incorporated it into our analysis.

---

> ### Author Response · Authors · 2024-11-29
> **Update Further Response about Language Bias in LLM-as-a-Judge**
>
> We have further discussed the issue of **Language Bias in LLM-as-a-Judge** in our approach with Reviewer `w67r`, and have updated the results of this discussion in the general response.
>
> In `Table 1 of the manuscript`, we evaluate the GPT score for individual responses in different languages, e.g., $y^{en}$ or $y^{ru}$. This approach may introduce language bias, as scores could vary depending on the language in which the response is generated. To mitigate this potential issue, we perform more detailed pairwise comparisons within the same language in `Tables 2 and 3 of the manuscript`, thereby avoiding cross-lingual scoring bias. Importantly, this language bias in `Table 1` does not affect our conclusion, as we focus on constructing pairwise comparisons between responses within the same language for `Tables 2 and 3`.
>
> Specifically, in each column of `Table 2`, both responses are in the same language. Take **'ru'** as an example. The **'Self Generation'** GPT score is calculated on $y^{ru}$ and the **'Self Translation'** GPT score is calculated on $y^{en \to ru}$. As both responses are in **'ru'**, it does not introduce potential language bias of LLM-as-a-Judge, providing a fair comparison. Similarly, in `Table 3`, the reward accuracy is evaluated on preference pairs consisting of $y^{ru}$ and $y^{en \to ru}$, which also avoids the cross-lingual comparison issue.
>
> Both `Tables 2 and 3` evaluate GPT score based on **pairwise comparisons within the same language**. This methodology inherently **avoids the cross-lingual comparison issue**, ensuring **a fairer and more consistent assessment** since all evaluations are conducted within the same language.
>
> **Summary:**
>
> - In `Table 1`, GPT scores are assigned to individual responses in different languages, this could introduce language bias. However, this does not affect our conclusion.
> - In `Tables 2 and 3`, pairwise comparisons are conducted **within the same language**, making the evaluation **fairer and more consistent**.
> - By translating responses into the same language, we create preference pairs that **avoid language-specific biases** and focus on the **relative quality** of the responses.
>
> Our approach, using the LLM-as-a-judge, ensures that the evaluation of responses is **consistent** and **unbiased**, as all comparisons are done within the **same language**.

---

### Meta-Review · Area_Chair_ToTs · 2024-12-22

**Metareview:**

This paper leverages DPO style optimization to improve multilingual instruction following and numerical reasoning in LLMs by leveraging the imbalance between dominant and non-dominant languages within LLMs is leveraged as a reward signal.

Strengths:
- A new method that shows gains in X-AlpacaEval leaderboard on the MGSM benchmark by training Meta-Llama-3-8B-Instruct.

Weaknesses:
- A reviewer mentions that this paper has high similarity with She et al. (2024), but I am not very concerned about this given the innovations that the authors mention clearly in one of their responses.
- Reviewers have concerns about using GPT as judge for evaluation, but I believe this was also addressed by the authors in their responses.

**Additional Comments On Reviewer Discussion:**

There was significant discussion between reviewers and authors for this paper, and two cases it resulted in increase in score.  I was not concerned by the issue raised by reviewer AMKf as I did not see an instance of plagiarism at all.

---

### Decision · Program_Chairs · 2025-01-22

Accept (Poster)